# High-throughput phenotyping of infection by diverse microsporidia species reveals a wild *C. elegans* strain with opposing resistance and susceptibility traits

**Calvin Mok**[1]*, **Meng A. Xiao**[1]ʘ, **Yin C. Wan**[1]ʘ, **Winnie Zhao**[1], **Shanzeh M. Ahmed**[1], **Robert J. Luallen**[2], **Aaron W. Reinke**[1]*

1 Department of Molecular Genetics, University of Toronto, Toronto, Ontario, Canada, 2 Department of Biology, San Diego State University, San Diego, California, United States of America

ʘ These authors contributed equally to this work.
* calvin.mok@utoronto.ca (CM); aaron.reinke@utoronto.ca (AWR)

**Data Availability Statement:** The sequence reads for all samples were submitted to the NCBI Sequence Read Archive under NCBI BioProject

## Abstract

Animals are under constant selective pressure from a myriad of diverse pathogens. Microsporidia are ubiquitous animal parasites, but the influence they exert on shaping animal genomes is mostly unknown. Using multiplexed competition assays, we measured the impact of four different species of microsporidia on 22 wild isolates of *Caenorhabditis elegans*. This resulted in the identification and confirmation of 13 strains with significantly altered population fitness profiles under infection conditions. One of these identified strains, JU1400, is sensitive to an epidermal-infecting species by lacking tolerance to infection. JU1400 is also resistant to an intestinal-infecting species and can specifically recognize and destroy this pathogen. Genetic mapping of JU1400 demonstrates that these two opposing phenotypes are caused by separate loci. Transcriptional analysis reveals the JU1400 sensitivity to epidermal microsporidia infection results in a response pattern that shares similarity to toxin-induced responses. In contrast, we do not observe JU1400 intestinal resistance being regulated at the transcriptional level. The transcriptional response to these four microsporidia species is conserved, with *C. elegans* strain-specific differences in potential immune genes. Together, our results show that phenotypic differences to microsporidia infection amongst *C. elegans* are common and that animals can evolve species-specific genetic interactions.

## Author summary

Animals display a large diversity of different characteristics within a population. One characteristic that varies within populations is susceptibility to infectious disease. Microsporidia are ubiquitous animal parasites that infect most types of animals, but the influence that these parasites have had on animal evolution and diversity is not well understood. Using wild isolates of the model organism *Caenorhabditis elegans*, we tested

PRJNA841603. Data from all the infection experiments is in S1 Data. PhenoMIP analysis scripts can be found at https://github.com/camok/PhenoMIP.

**Funding:** This work was supported by Canadian Institutes of Health Research grants no. 400784 and 461807 and an Alfred P. Sloan Research Fellowship FG2019-12040 to A.W.R. and by an NIH grant GM146836 to R.J.L. The funders had no role in study design, data collection and analysis, decision to publish, or preparation of the manuscript.

**Competing interests:** The authors declare that they have no competing interests.

the diversity of responses to infection by four naturally occurring microsporidia species. These experiments were done with a pooled assay that allowed for the determination of the relative abundance of strains within a population under different infection conditions. These experiments identified a *C. elegans* strain that is sensitive to infection by one species of microsporidia but resistant to another. Genetic mapping experiments show that different genetic regions are responsible for these different infection responses. Together our study demonstrates that there is variability in the susceptibility of *C. elegans* to microsporidia infection and suggests that these parasites have likely had a large influence on animal evolution.

## Introduction

Obligate intracellular parasites only propagate inside of host cells, which places extreme pressure on the host, resulting in genomic changes [1–3]. Animals have coevolved with pathogens and genetic variation to pathogen susceptibility is common in animal populations [4,5]. Much of this genetic and phenotypic diversity to infection is specific to individual species of pathogens or even to different strains [6,7]. Hosts can evolve several ways to improve their fitness in the presence of pathogens. They can become more resistant by preventing pathogen invasion or by limiting pathogen growth [8]. Alternatively, hosts can evolve to better tolerate the infection, by limiting the negative effects of infection without reducing the amount of pathogen [9,10].

Microsporidia are a large phylum of eukaryotic obligate intracellular pathogens with a wide host range including humans and agriculturally important animals such as honey bees and shrimp [11–14]. Microsporidia infections cause developmental delays, a reduction in progeny, and increased mortality [15–17]. Several studies have shown that microsporidia can affect host population dynamics and evolution. Microsporidia have been suggested to influence the composition of *Caenorhabditis elegans* genotypes and resistant strains can outcompete sensitive ones in just a few generations [18–20]. In competition experiments with over a dozen isolates of *Daphnia magna*, different microsporidia species have been shown to select for specific strains [21,22]. Using genetic mapping, several genomic regions contributing to microsporidia resistance in *C. elegans*, *D. magna*, and honey bees have been identified [19,23–26].

*C. elegans* has become a powerful model for studying host-pathogen coevolution [27–30]. In the wild, *C. elegans* contend with bacterial, fungal, viral, and microsporidian pathogens and these coevolving species represent strong selective pressures on the animal's genome [31,32]. To facilitate the study of natural diversity, hundreds of isolates of *C. elegans* have been collected from around the world [33]. These wild isolates have different levels of susceptibility to pathogens and natural variants have been discovered that impact traits such as bacterial avoidance and viral susceptibility [7,19,34–37]. Although studying phenotypic variation in wild isolates is a powerful approach, measuring each strain individually is time consuming and laborious. To circumvent this issue, strategies have been developed to perform multiplex competitive fitness assays under selective conditions such as starvation stress or different bacterial diets [38–40]. Several sequencing-based approaches, termed either PhenoMIP or MIP-seq, determine the identity of these individual strains by using molecular inversion probes (MIPs) to enrich for their unique genomic signatures [39,40].

A common microbial infection found in wild *C. elegans* is from species of microsporidia in the *Nematocida* genus. Infection of *C. elegans* by microsporidia begins with ingestion of a chitin-coated spore [41]. The spore then fires its unique infectious apparatus, known as a polar

tube, and deposits its sporoplasm within the host [42]. The sporoplasm then replicates to form membrane-bound, multinucleated cells know as meronts which then differentiate into spores that are shed by the host [16]. Some *Nematocida* species, such as *N. parisii*, *N. ausubeli*, and *N. ironsii*, infect only the intestine, while others, such as *N. displodere*, *N. ferruginous*, and *N. cider*, can proliferate in several tissues including the muscle and epidermis [15,19,41,43,44]. *C. elegans* possesses several mechanisms to defend itself against microsporidia [45,46]. Infection with microsporidia induces transcriptional upregulation of genes known as the Intracellular Pathogen Response (IPR) [47]. Mutations in several negative regulators of the IPR cause elevated expression of the IPR and resistance to microsporidia infection [48–50]. *C. elegans* also possess a separate form of inherited immunity that is activated by infection of parents and is transferred to progeny [49].

Here we present the results of a high-throughput fitness competition of 22 pooled *C. elegans* wild isolates infected by four species of microsporidia at multiple doses. We identified 16 candidate interactions of either host resistance or sensitivity. We demonstrate that the wild isolate JU1400 has distinct and opposing responses to infection by different microsporidia species. This strain has a defect in tolerance to the epidermally-infecting *N. ferruginous* but is resistant to the intestinal-infecting *N. ironsii* through specific elimination of infection by this species. Through mapping experiments and the generation of recombinant strains, we demonstrate that the opposing phenotypes of JU1400 are caused by separate loci. In contrast to the species-specific phenotypes we observed from fitness competitions, transcriptional analysis showed a host response to infection that is mostly independent of the microsporidia species, but dependent upon the *C. elegans* strain. Notably, we identified a signature of downregulated genes that has overlap with the *C. elegans* response to toxins, and this downregulated response was most prominent in the tolerance-defective strain JU1400 infected with *N. ferruginous*. Our results demonstrate that PhenoMIP can efficiently identify resistant and sensitive wild isolates and that microsporidia genotype-specific interactions are common in *C. elegans*.

## Results

### A PhenoMIP screen of wild isolate strains reveals diverse responses to microsporidia infection

To identify *C. elegans* strains that vary in their resistance or susceptibility to microsporidia infection, we carried out a modified PhenoMIP screen. We chose 22 *C. elegans* wild isolate strains that are genetically diverse and isolated from a range of geographic regions (**S1 Fig**). We selected four pathogenic microsporidia species that differ in their impact on *C. elegans* development and the tissues they infect (*N. parisii*, *N. ausubeli*, and *N. ironsii* infect the intestine and *N. ferruginous* infects the muscle and epidermis) (**Fig 1A and S1 Table**) [15,16,19,43,51]. To synchronize infections, we treated adult populations with a sodium hydroxide and sodium hypochlorite mixture (known as bleach synchronizing) and allowed the recovered embryos to hatch in the absence of food. The process of bleach synchronizing animals for microsporidia infection has been shown to have similar levels of infection between bleached versus laid embryos, and short-term starvation is reported to have minimal impact on wild isolates [19,20,40]. We pooled equal numbers of synchronized first larval stage (L1) wild isolates and a laboratory reference-derived strain (VC20019) into a single population. VC20019, was chosen as a reference because it has many genomic variants that can be used to distinguish it from N2 while having developmental timing and fecundity similar to that of N2 [39,52]. We then plated these animals along with their food source of *Escherichia coli* bacteria mixed with one of four doses of each microsporidia species (high, medium, low, and none) (**S2**

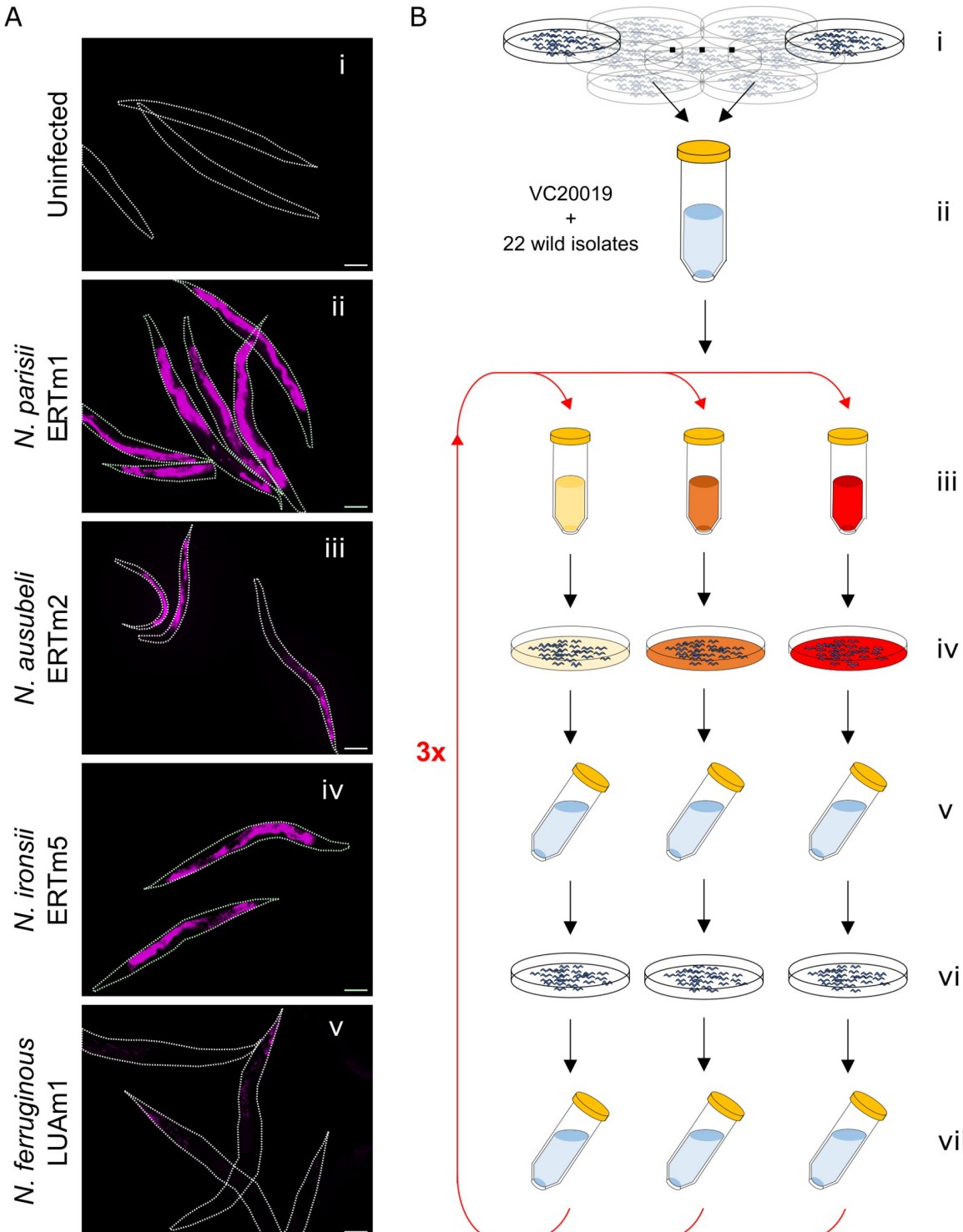

**Fig 1. A PhenoMIP high-throughput screen to identify wild strains that are either resistant or susceptible to microsporidia infection.**
(A) N2 L1 animals that were uninfected (i) or exposed to either 3.5 million *N. parisii*, *N. ironsii*, or *N. ausubeli* spores or 1.25 million *N. ferruginous* spores for 72 hours, were fixed and stained with the following 18S FISH probes (i, ii, iv; MicroB-CAL Fluor Red 610, iii; Microsp1A-FAM and v; a combination of three Fluor Red 610 probes described in the Methods). Scale bars, 100 μm. (B) PhenoMIP screening protocol for microsporidia requires that separate unstarved populations for each of the 22 wild isolate strains as well as VC20019 (lab reference-derived strain) be sodium hyplochlorite-synchronized (i) before pooling in equimolar amounts (ii) and infecting (iii) with

different microsporidia species at doses that can impose fitness challenges to the lab reference strain. Populations are grown for 72 hours post-infection (iv) before sodium hypochlorite-synchronizing (v) and growing free of infection for an additional 72 hours (vi). Uninfected populations are harvested, subsampled for genomic DNA, and sodium hypochlorite-synchronized (vii) before repeating the infection and rest phases for an additional three rounds.

Table). We hypothesized that a high-dose exposure could identify *C. elegans* strains that were resistant or tolerant to microsporidia infection while medium- and low-dose exposures might identify strains exhibiting mild to strong susceptibility to microsporidia infection. Animals were exposed to microsporidia spores for 72 hours at 21˚C, before collecting and bleach synchronizing the population. This mixture inhibits microsporidia from infecting the next generation of animals and synchronizes the population [49]. This microsporidia-free population of L1s was then grown in the absence of infection for 72 hours at 21˚C to allow intergenerational immunity to reset [49]. Except for the *N. ausubeli* medium-dose infection replicates (**see Methods**), worms in each condition were infected for an additional three cycles for a total of eight generations. Adult animals were collected every two generations at the end of the uninfected phase and genomic DNA was isolated for further analysis (**Fig 1B**).

To determine how population composition changes due to exposure to microsporidia, we calculated the relative abundance of each strain within each condition. We used MIPs to target strain-specific variants for sequencing at high depth. In total we sequenced 127 samples, which is the equivalent of 2162 infections and 736 mock/control infections. From each of the eight mock-infection controls and twenty-four infection replicates for every strain we calculated a mean fold-change rate (FCR) from across all timepoints. The distribution of FCRs for each strain was then converted to a modified z-score distribution to identify candidate interactions (**See Methods**). From this dataset, we identified 16 nematode/microsporidia pairings where the modified z-score for both high-infection replicates was consistently outside the range of +/- 1.5 (**Figs 2 and S2 and Table 1**). We also generated mean growth profiles across all timepoints for each worm strain in environments that were either pathogen-free or infected by each of the microsporidia species (**Fig 3A and 3B**). From our analyses we observed ten interaction pairs with reduced population fitness (low-fitness strains), suggesting potential susceptibility to infection. In addition, we observed six interactions with increased population fitness (high-fitness strains), suggesting a potential resistance to infection conditions. Our high-fitness interaction set included the previously reported resistance interaction of CB4856 to *N. ironsii* (**S3 Fig**) [19,51] which further suggested our scoring threshold was appropriate for the detection of interactions in this assay format. Of the 15 novel candidate interactions, we identified five interactions with *N. ironsii* (ERTm5), eight with *N. ferruginous* (LUAm1), two with *N. ausubeli* (ERTm2), and no strong interactions with *N. parisii* (ERTm1). In total, our screen yielded 13 *C. elegans* wild isolate strains with significant interactions, including the strains JU360, JU1400, and MY2 that had interactions with more than one species of microsporidia (**Fig 2 and Table 1**).

## JU1400 fails to tolerate *N. ferruginous* infection

To validate a subset of our large-scale multiplex identification of microsporidia-strain phenotypes, we performed isogenic population infection experiments. We infected individual strains with doses of microsporidia spores equivalent to those used in the PhenoMIP experiment (**Figs 3C and 3D and S4–S6**). We examined a combination of phenotypes including morphology of adult populations, formation of microsporidia meronts or spores, and embryo production. Most high-fitness strains tended to experience a smaller reduction in embryo production ratios compared to changes in lab reference N2 in identical infection conditions. Low-fitness

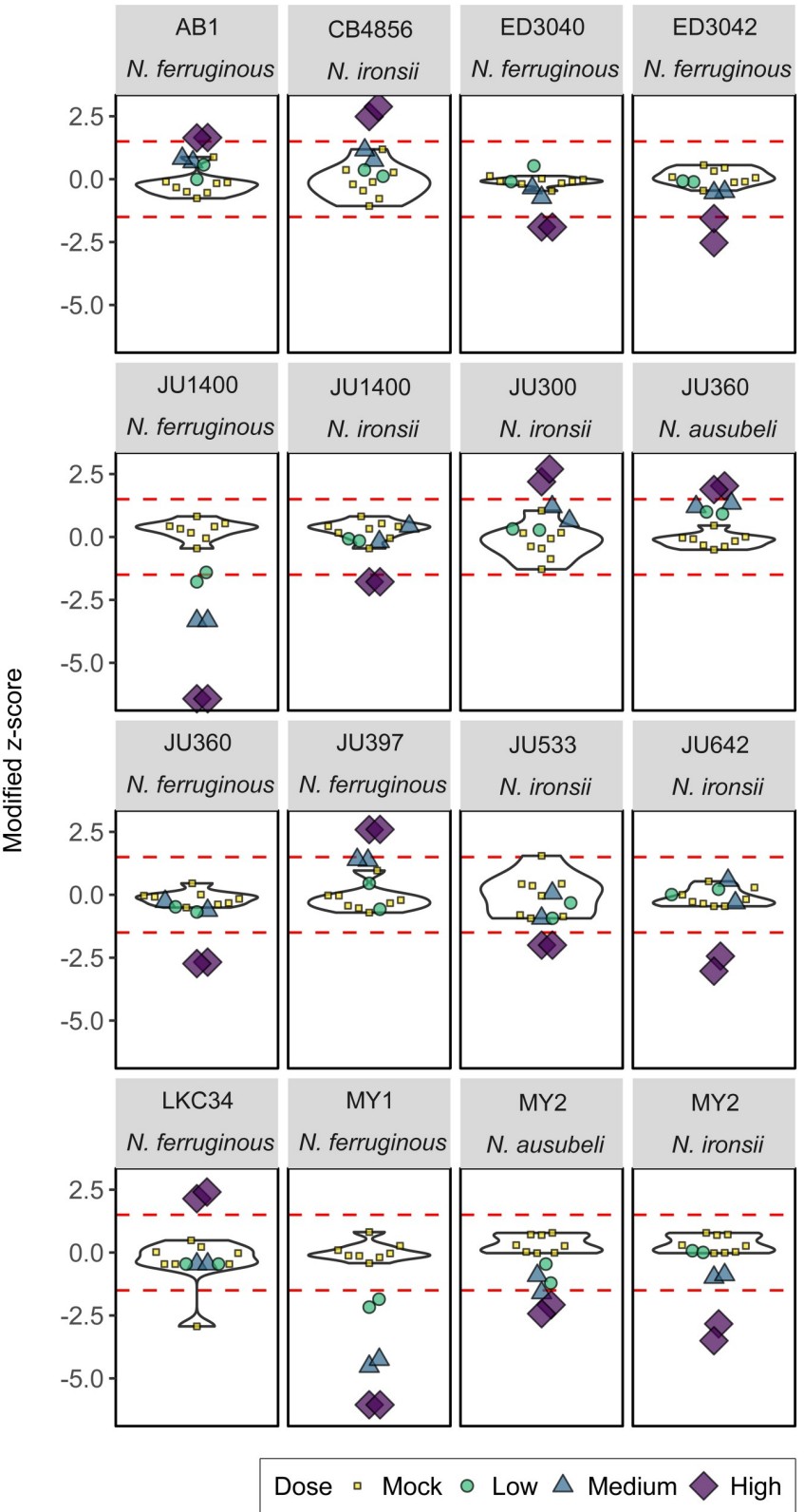

**Fig 2. Analysis of PhenoMIP mean fold-change rate identifies 16 candidate nematode/microsporidia interactions.**
Violin plots of *C. elegans* wild isolate strains infected by specific microsporidia species. Violins are derived from

modified z-scores of mean fold-change rate (FCR) per generation using the eight mock-infection replicates of that strain. Overlayed are the six infection replicates of low (green circles), medium (blue triangles), and high (purple diamonds) doses for the microsporidia species indicated in each subplot title. Mock infection replicates are also overlayed (yellow squares). Red dotted lines represent the candidate hit threshold for modified z-scores of ±1.5.

strains, on the other hand, experienced developmental delay phenotypes and/or a reduction in fecundity–both of which would contribute negatively to the fitness readout observed via Phe-noMIP. These phenotypes were most evident in JU1400 and MY1 which both had severe, dose-dependent responses to infection by the epidermal microsporidia species *N. ferruginous* (**Figs 3C and 3D and S7**). In total, of the 12 pairs that we observed (**Table 1**), nine had altered relative fecundity profiles that could account for the fitness changes observed from our Pheno-MIP screen. Additionally, of the seven pairs we retested that had a z-score below -1.5, six had a significant reduction in embryos when infected, compared to N2. In summary, we confirmed that 75–87.5% (depending on the z-score cut off) of the strains that we tested individually have the phenotype identified in our PhenoMIP assays.

To examine how a *C. elegans* strain responds to different microsporidia species, we focused on JU1400, in which we observed reduced fitness to infection by both *N. ferruginous* and *N. ironsii*. To determine why JU1400 appeared sensitive to *N. ferruginous*, we performed pulse-chase experiments by infecting animals for 3 hours, washing away free spores, and replating animals for an additional 69 hours [19]. We observed that the percentage of JU1400 and N2 animals infected with *N. ferruginous* was similar (**Figs 3E and S8A and S8B**). Additionally, the amount of meronts formed during infection did not significantly differ between N2 and JU1400 (**Figs 3F and S8C and S8D**). Taken together, these results demonstrate that although

**Table 1. PhenoMIP screen interaction hits.**

| Strain | Site of isolation | Microsporidia species | Relative Impact on host fitness | Modified z-score | Single infection embryo phenotype? |
|--------|-------------------|----------------------|--------------------------------|------------------|-----------------------------------|
| AB1 | Adelaide, Australia | *N. ferruginous* | Increase | 1.65, 1.64 | Yes |
| JU397 | Ouistreham, France | *N. ferruginous* | Increase | 2.58, 2.59 | Yes |
| LKC34 | Antananarivo, Madagascar | *N. ferruginous* | Increase | 2.14, 2.40 | Untested |
| ED3040 | Johannesburg, South Africa | *N. ferruginous* | Decrease | -1.90, -1.90 | Untested |
| ED3042 | Capetown, South Africa | *N. ferruginous* | Decrease | -2.53, -1.57 | Yes |
| JU360 | Franconville, France | *N. ferruginous* | Decrease | -2.68, -2.74 | Yes |
| MY1 | Munster, Germany | *N. ferruginous* | Decrease | -6.05, -6.05 -4.52[M], -4.25[M] -2.17[L] -1.85 [L] | Yes |
| JU1400 | Sevilla, Spain | *N. ferruginous* | Decrease | -6.43, -6.43 -3.33[M], -3.33[M] | Yes |
| CB4856 | Hawaii, USA | *N. ironsii* | Increase | 2.48, 2.88 | Yes |
| JU300 | Le Blanc, France | *N. ironsii* | Increase | 2.19, 2.70 | No |
| MY2 | Munster, Germany | *N. ironsii* | Decrease | -2.83, -3.50 | Yes |
| JU533 | Primel-Tregastel, France | *N. ironsii* | Decrease | -2.00, -2.00 | Untested |
| JU642 | Paris, France | *N. ironsii* | Decrease | -2.44, -3.06 | Untested |
| JU1400 | Sevilla, Spain | *N. ironsii* | Decrease | -1.78, -1.78 | Yes |
| JU360 | Franconville, France | *N. ausubeli* | Increase | 2.03, 1.88 | No |
| MY2 | Munster, Germany | *N. ausubeli* | Decrease | -2.07, -2.42 | No |

All modified z-scores are identified from high dose infections unless otherwise noted.

[M] Medium dose infection modified z-score

[L] Low dose infection modified z-score

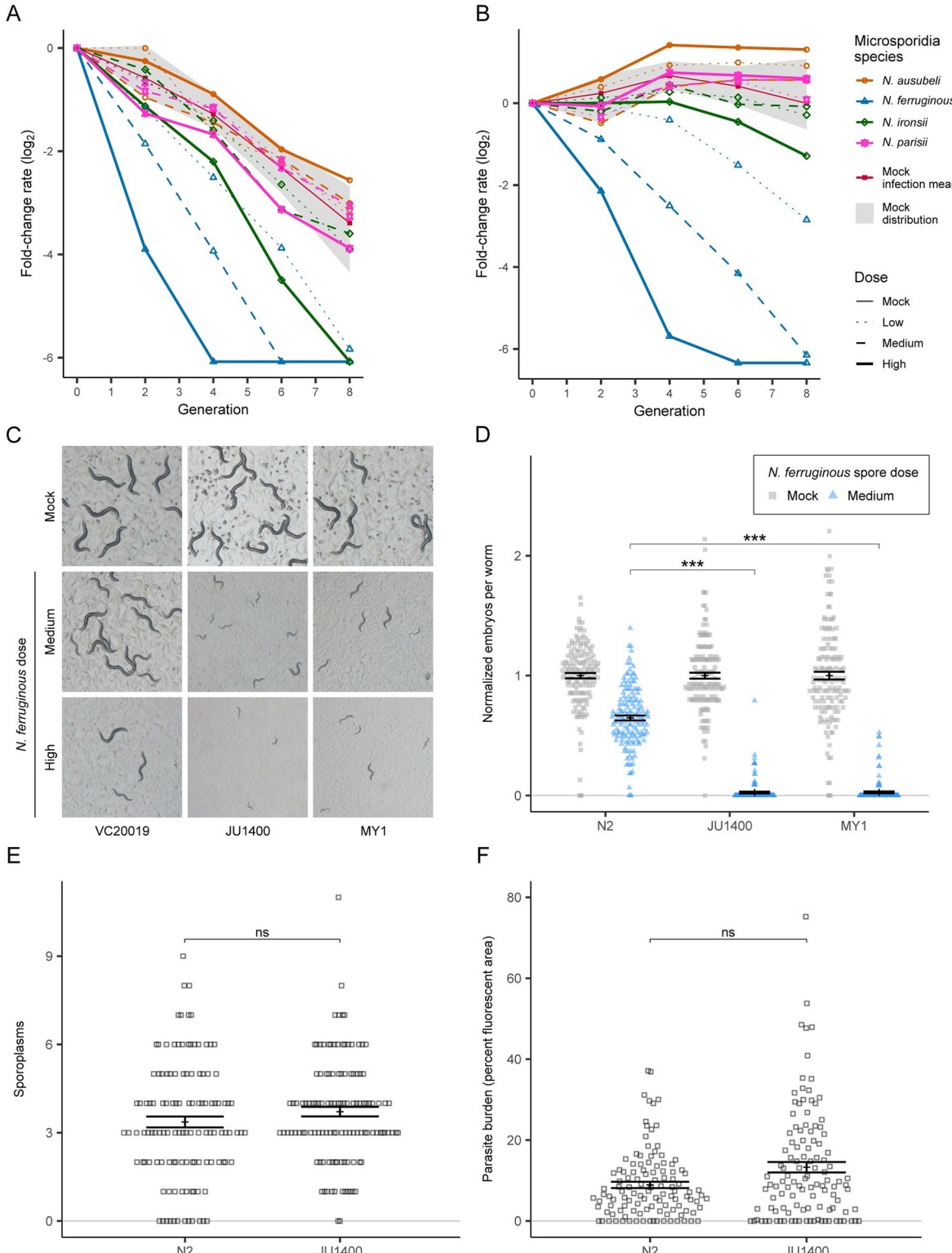

**Fig 3. JU1400 lacks tolerance to *N. ferruginous* infection.** (A-B) Line graph of fold change across multiple time points and conditions for JU1400 (A) and MY1 (B). Microsporidia strains are denoted by point shape and doses of microsporidia are denoted by line type. The eight uninfected control replicates are bounded within the greyed area and the mean of the uninfected control is denoted by a solid red line. All other lines represent mean FCR of two biological replicate samples. (C-D) Strains were infected for 72 hours with *N. ferruginous*. (C) Dissecting microscope images of live animals growing on plates. (D) Dot plot of normalized embryo counts of animals fixed and stained

with DY96 with mean and SEM bars. Data is combined from three biological replicates with n = {41,50} worms per replicate sample. (E-F) N2 and JU1400 animals were infected for three hours with 1.25 million spores of *N. ferruginous* before washing away uneaten spores from each population. Animals were fixed after 3 hours (E) or after 72 hours (F). (E) Dot plot of sporoplasms counted in N2 and JU1400 animals with mean and SEM bars. Data is combined from three biological replicates with n = 40 worms per replicate sample. (F) Dot plot with SEM bars depicting mean parasite burden as a function of percent fluorescent area per animal by FISH staining in N2 and JU1400 populations. Data is combined from three biological replicates with n = {29, 53} worms per replicate sample. p-values were determined by two-way ANOVA with Tukey post-hoc. Significance was defined as p ≤ 0.001 (***) and not significant as p > 0.05 (ns).

JU1400 is more sensitive to the effects of *N. ferruginous*, JU1400 is not significantly more infected, suggesting a defect in tolerance.

To determine the specificity of JU1400 to *N. ferruginous* infection, we infected JU1400 with other related microsporidia species and strains. We infected JU1400 with a second isolate of *N. ferruginous*, LUAm3, as well as a separate muscle and epidermal-infecting microsporidian, *N. cider* [44]. Infection by LUAm3 caused the same severe phenotype as LUAm1 (**S9A and S9B Fig**). JU1400 is infected by *N. cider* to a similar extent as N2, but infection by this species resulted in a dose-dependent reduction in embryo numbers in JU1400 that was not observed in N2 (**S9C and S9D Fig**).

## JU1400 specifically clears *N. ironsii* infection

In addition to the verified susceptibility to *N. ferruginous*, our PhenoMIP screen also identified a more moderate fitness disadvantage for JU1400 in high-dose exposures to *N. ironsii* (**Fig 3A**). We infected N2 and JU1400 with *N. ironsii* for 72 hours and compared the relative change in embryos for each strain versus mock-infected controls. We observed no significant difference in this normalized number of embryos between the N2 and JU1400 strains (**Fig 4A**). However, JU1400 is significantly less infected, suggesting that JU1400 is resistant to *N. ironsii* infection (**Figs 4C and S10A and S10C**). To determine whether the observed resistance conferred a fitness advantage at a later time point, we exposed JU1400 animals to *N. ironsii* for 96 hours and observed significantly more embryos and significantly less infection compared to N2 (**Figs 4B and 4D and S10B and S10D**). Together our data suggest that although JU1400 appears to have a short-term disadvantage in our pooled competition assays, this strain has a long-term advantage due to enhanced resistance towards *N. ironsii*.

Several mechanisms of *C. elegans* resistance to microsporidia infection have been described, including a block of invasion or clearance of infected parasite [19,20,49,51]. To determine how JU1400 is resistant to *N. ironsii* infection, we performed a pulse-chase experiment by infecting animals for 3 hours, washing away any external microsporidia spores, and examining animals at 3, 24, and 72 hours post infection (hpi). Both N2 and JU1400 had similar numbers of sporoplasms at 3 hpi, but JU1400 had decreased levels of pathogen at both 24 and 72 hpi (**Fig 5A**). This experiment revealed that JU1400 does not block microsporidia invasion, but instead has the capability to actively eliminate *N. ironsii* infection. To determine if the resistance in JU1400 was dependent upon development stage, we performed pulse-chase infections at the L4 stage and observed that JU1400 animals could no longer clear the infection (**Fig 5B**).

Resistance to *N. ironsii* has previously been observed in the *C. elegans* wild isolate, CB4856 [19]. This resistance was shown to occur through a modest decrease in invasion coupled with the ability to clear the infection at the L1 stage. To compare the properties of these two resistant strains, we infected both strains with *N. ironsii* for 72 hours. This experiment shows that JU1400 is more resistant to infection (**Fig 5C**). To determine the specificity of JU1400 resistance, we tested infection with either *N. parisii* or *N. ausubeli* for 72 hours. CB4856 was slightly, but not significantly less infected by *N. parisii* and *N. ausubeli*, whereas JU1400 was infected by these species to the same extent as N2 (**Fig 5D**). We also tested the ability of

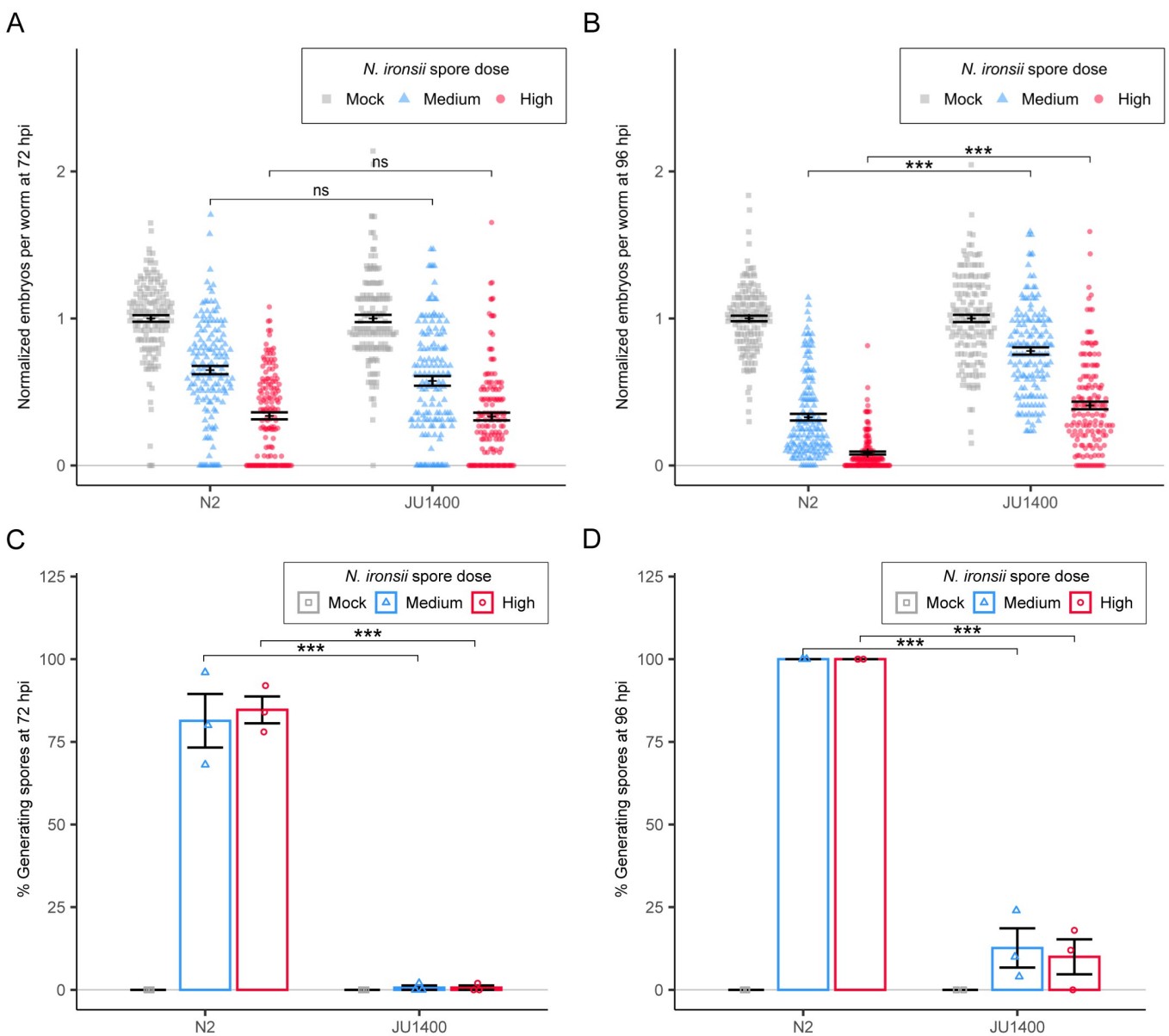

**Fig 4. JU1400 is resistant to *N. ironsii* infection.** (A-D) N2 and JU1400 were infected with *N. ironsii* for 72 hours (A and C) or 96 hours (B and D), fixed, and stained with DY96. (A-B) Dot plots of normalized embryo counts of animals fixed and stained with DY96 with mean and SEM bars. Data is combined from three biological replicates with n = {47,50} worms per replicate sample. (C-D) Bar plot with SEM bars of percent population with new spore formation from three combined replicate experiments. p-values were determined by two-way ANOVA with Tukey post-hoc. Significance was defined as p ≤ 0.001 (***) and not significant as p > 0.05 (ns).

JU1400 and CB4856 to clear microsporidia infection, and both strains could only eliminate *N. ironsii* (**S10E and S10F Fig**). Together, these results demonstrate that JU1400 resistance is specific to *N. ironsii*.

The immunity of JU1400 against *N. ironsii* could be a specific recognition of *N. ironsii* infection that induces broad immunity towards microsporidia or the immunity itself could be specific for *N. ironsii*. To distinguish between these possibilities, we set up coinfection experiments using species-specific FISH probes and then coinfected JU1400 and N2 animals with *N.*

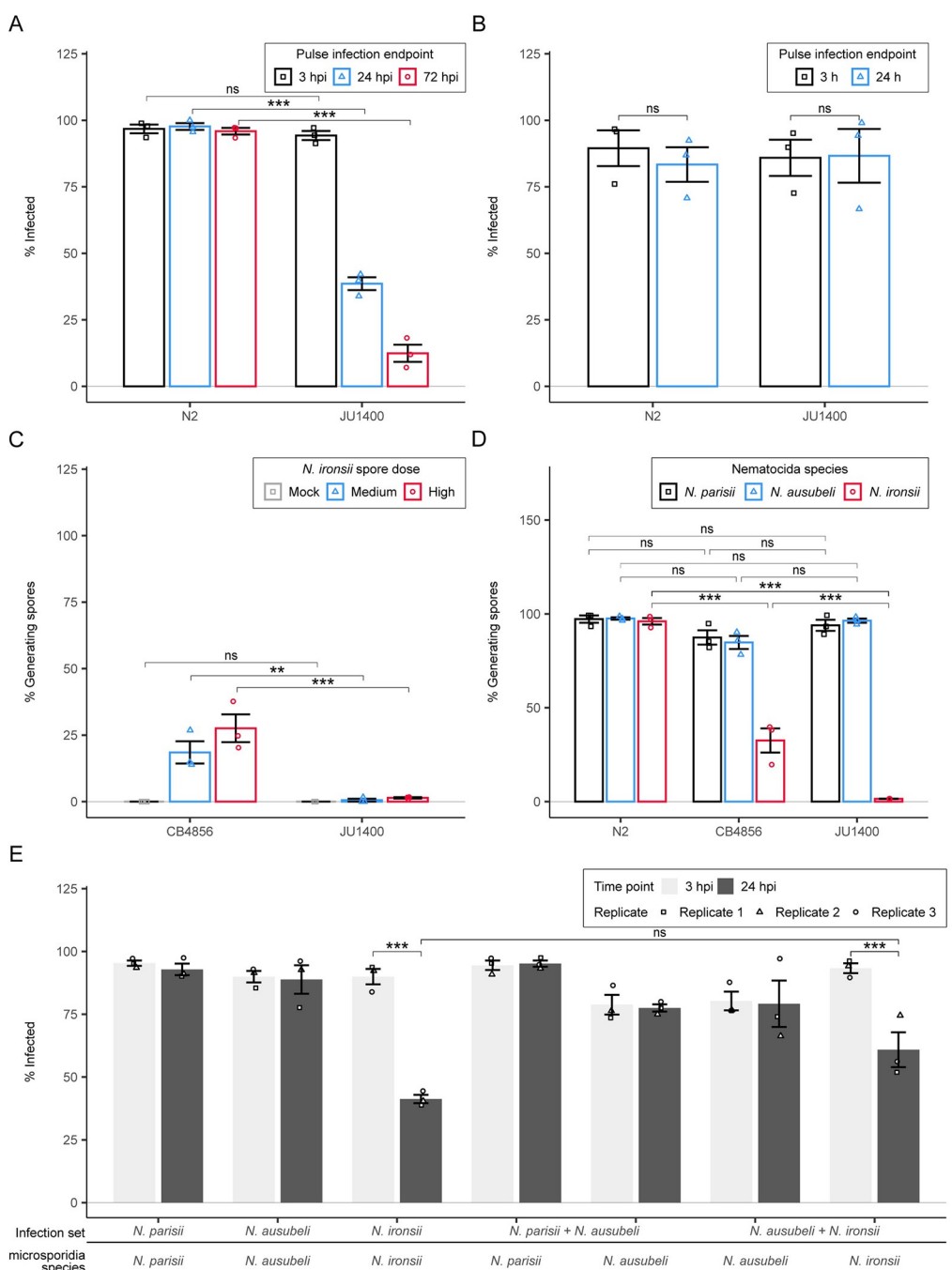

**Fig 5. JU1400 specifically clears *N. ironsii* infection at the earliest larval stage.** (A-B) N2 and JU1400 animals were pulse infected for 3 hours with *N. ironsii* before washing away excess spores. Animals were pulse infected at either the L1 stage and fixed at 3-, 24-, and 72 hpi (A) or animals were pulse infected at the L4 stage before fixing at 3- and 24 hpi (B). Samples were then stained with an *N. parisii* 18S RNA FISH probe and animals containing either sporoplasms or meronts were counted as infected. Bar plot with SEM bars show percent of animals infected per replicate from three biological replicates with n = 20 worms per replicate. (C) CB4856 and JU1400 were infected at the L1 stage with either medium or high doses of *N. ironsii* for 72 hours and then fixed and stained with DY96. (D) N2, JU1400, and CB4856 were infected at the L1 stage with either *N. parisii*, *N. ausubeli*, or *N. ironsii* for 72 hours and then fixed and stained with DY96. (C-D) Bar plot with SEM bars show percent of animals generating spores per replicate from three biological replicates with n = 20 worms per replicate. (E) JU1400 animals were pulse infected for 3 hours with *N. parisii*, *N. ausubeli*, *N. ironsii*, *N. parisii* + *N. ausubeli*, or *N. ausubeli* + *N. ironsii* before washing away excess spores. Animals were either fixed immediately at 3 hpi or at 24 hpi. Samples were then stained with species-specific 18S rRNA FISH probes and animals containing either sporoplasms or meronts were counted as infected. Bar plot with SEM bars

show percent of animals infected per replicate from three biological replicates with n = {78, 147} worms per replicate. p-values were determined by two-way ANOVA with Tukey post-hoc. Significance was defined as p ≤ 0.01 (**), p ≤ 0.001 (***), and not significant as p > 0.05 (ns).

*ausubeli* and either *N. ironsii* or *N. parisii*. Under these conditions, *N. ironsii* sporoplasms are eliminated regardless of the presence of *N. ausubeli*. There is no clearance of *N. ausubeli* observed, either by itself or in the presence of *N. ironsii*. (**Figs 5E** and **S11A and S11B**). No clearance was observed in N2 regardless of the infection condition (**S11C Fig**). These results further demonstrate that JU1400 has specific immunity towards *N. ironsii*.

## Genetic mapping reveals multiple loci are responsible for the interactions of JU1400 with microsporidia

The opposing phenotypes of JU1400 in response to microsporidia infection could be caused by the same or different genetic loci. To determine the genetic basis for JU1400 being sensitive to *N. ferruginous* and resistant to *N. ironsii*, we mapped the causative alleles using a pooled competitive fitness assay [52]. We crossed JU1400 to the mapping strain VC20019 and infected F2 animals with either *N. ferruginous* for 72 hpi or *N. ironsii* for 96 hpi (**S3 Table** and **Methods**). After two rounds of selection, genomic DNA was isolated from each sample and MIP--MAP sequencing libraries were generated [52]. As a control, we used this approach to map the resistance of CB4856 to *N. ironsii*. We then compared our results to a previous study that determined regions responsible for resistance [19]. Although the previous mapping study was based on pathogen load as opposed to our mapping of competitive fitness advantage, several regions mapped concordantly between the two approaches on chromosome II and the right arm of chromosome V (**S12 Fig**).

Our mapping results of JU1400 with *N. ferruginous* identified a single peak of VC20019 marker fixation on the left-hand side of chromosome I (**Fig 6A**). This mapping was repeated with MY1, another strain that exhibited a fitness disadvantage with *N. ferruginous* (**Fig 3B–3D**). The mapped region under negative selection was the same in MY1. The MY1 strain also carries the *zeel-1/peel-1* incompatibility locus at chromosome I, which accounts for the specific signal seen at the I:2,726,021 marker in both the infected and uninfected samples (**S13A Fig**) [53]. To determine if the same locus was responsible for *N. ferruginous* in MY1 and JU1400, we tested their ability to complement. First, we determined that both MY1 and JU1400 display a recessive phenotype when crossed to our control strain. We then mated MY1 and JU1400 and the resulting cross progeny were susceptible to *N. ferruginous* infection, indicating that the two strains do not complement (**S14A and S14B Fig**). We then infected MY1 with *N. ironsii*. There were fewer MY1 animals containing spores compared to N2, and less JU1400 animals had spores or meronts compared to MY1 animals (**S14C and S14D Fig**). Together this data suggests a region on the left side of chromosome I, is responsible for sensitivity to *N. ferruginous* in multiple strain*s*.

Our mapping of JU1400 with *N. ironsii* at 96 hours identified three regions that are linked with resistance to infection, one on the left-hand side of chromosome II from 1,639,865–4,426,687, a second in a broad area of chromosome V from 11,035,658–18,400,066, and a third on the right-hand side of chromosome X from 15,180,772–17,676,467 (**Fig 6B**). A similar pattern was observed when these mapping experiments were performed at 72 hours, with the exception that the region on chromosome X was no longer observed (**S13B Fig**). We also observed a region on the left side of chromosome I where JU1400 variants underwent weak negative selection upon multiple *N. ironsii* infection rounds. Together, our data suggests that JU1400 resistance to *N. ironsii* is multigenic and may be influenced by infection time.

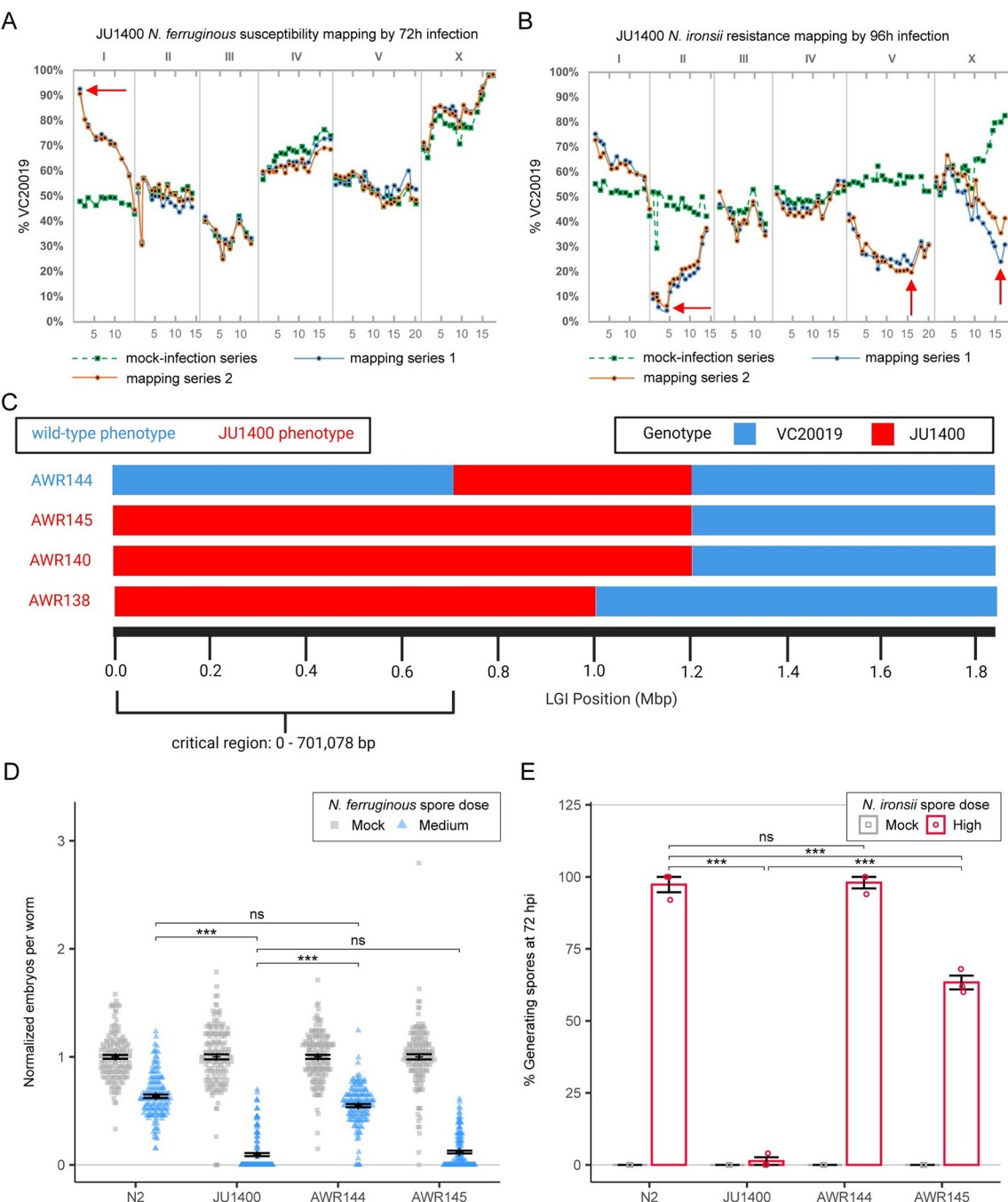

**Fig 6. Genetic mapping of JU1400 host-response phenotypes identifies distinct loci.** Genetic mapping of JU1400 susceptibility to *N. ferruginous* (A) and JU1400 resistance to *N. ironsii* (B) was performed with MIP-MAP using the AWR133 mapping strain. Green square-points and dotted lines represent a mock-infection replicate with the solid blue and orange lines representing two biological replicates exposed to specific infection conditions. Red arrows indicate candidate regions of interest based on the expected genome fixation direction during phenotype selection. (C) Whole-genome sequencing haplotype map of chromosome I NILs generated from crosses between JU1400 and AWR133. Regions of chromosome I for specific NILs are coloured by their inferred haplotype. Strain names are indicated to the left of each with AWR138, AWR140, and AWR145 sharing a susceptibility phenotype (Red text) while AWR144 is not susceptible (Blue text) to *N. ferruginous* infection. (D) N2, JU1400, AWR144, and AWR145 L1 stage animals were infected with *N. ferruginous* before being fixed at 72 hpi and stained with DY96 to visualize embryos. Dot plot of normalized embryo counts with mean and SEM bars. Data is combined from three biological replicates with n = 50 worms per replicate sample. (E) N2, JU1400, AWR144, and AWR145 L1 stage animals were infected with *N. ironsii* before being fixed at 72 hpi and stained with DY96 to visualize mature spores. Bar plot with SEM bars show percent of animals generating new spores per replicate from three biological replicates with n = 50 worms per replicate. p-values were determined by two-way ANOVA with Tukey post-hoc. Significance was defined as p ≤ 0.001 (***) and not significant as p > 0.05 (ns).

To confirm that the region on chromosome I was responsible for sensitivity to *N. ferruginous* infection, we generated Near Isogenic Lines (NILs). These NILs were constructed by crossing a version of VC20019 that contained a CRISPR-integrated single-copy GFP on chromosome I (AWR133) (**S15A Fig**). We tested ten independent NILs that contained at least some JU1400 sequence to the left of I:2,851,000. All ten of these NILs were sensitive to infection by *N. ferruginous* (**S15B Fig**). We used molecular markers to genotype these NILs, mapping the susceptibility locus to I:1–1,051,810 (**S4 Table**). We then outcrossed one of these NILs, AWR140 back to N2 six times. During this outcrossing process, we recovered two lines, AWR144 and AWR145, that contained different responses to *N. ferruginous* infection (**Fig 6D**). We performed whole-genome sequencing of AWR138 (another independently isolated NIL), AWR140, AWR144, and AWR145 which narrowed the critical region of association for our phenotype to I:1–701,078 (**Fig 6C**). Within this region, we identified 12 genes shared between JU1400 and MY1 (same complementation group) but absent in ED3042 and JU360 which tested into separate complementation groups (**S5 Table**). The NIL AWR145 is sensitive to *N. ferruginous* infection but is only partially resistant to infection by *N. ironsii* (**Fig 6D and 6E**). Together, our data demonstrate that the *N. ferruginous* sensitivity and *N. ironsii* resistance to JU1400 are determined by distinct genetic loci.

## Transcriptional analysis of the JU1400 response to microsporidia infection

To determine how the JU1400 strain differs in its response to microsporidia infection, we performed transcriptional analysis. We grew three biological replicates of N2 and JU1400 L1 stage animals either without microsporidia or infected with one of *N. parisii*, *N. ironsii*, *N. ferruginous*, or *N. ausubeli* for 48 hours. We then extracted and sequenced the *C. elegans* and microsporidia mRNA. First, we mapped the microsporidia mRNA in each of the eight infected samples. For the three intestinal infecting species, between ~1–9% of reads were mapped for each sample, with the exception of *N. ironsii* in JU1400, where few reads were identified, which is consistent with this strain being resistant to this species. For *N. ferruginous*, fewer reads were observed in either strain compared to the intestinal species, which is consistent with what we observed by microscopy (**S16 Fig** and **S6 Table**).

We then sought to determine how the two *C. elegans* strains responded to the four different microsporidia species (**S7 Table**). First, we used principal component analysis (PCA), and projected samples back onto the two principal components which accounted for more than 80% of the variance in our samples. PCA analysis revealed a clear clustering between the two strains (**Fig 7A**) and a secondary analysis by strain clustered the uninfected samples against pathogen infected samples (**S17 Fig**). Second, we clustered the differential expression profiles of the eight infected samples. This approach revealed several large groups of genes that were differentially expressed either across most samples or within the same strain (**Fig 7B** and **S8 Table**). Although the largest difference in the samples is between the two strains, we observe significant correlation between strains infected with each microsporidia species (**S18 Fig**).

We then determined the properties of genes that are differentially regulated in each sample. The number of differentially upregulated or downregulated genes ranged from 6–819, with the most being in JU1400 infected with *N. ferruginous* (**Fig 7C**). We then classified differentially expressed genes into either conserved, microsporidia species (pathogen) specific, *C. elegans* strain specific, or strain-pathogen specific categories (**Fig 7D**). We identified 97 genes upregulated in all samples, and no genes that were broadly downregulated. We identified ~30–50 differentially expressed genes specific to either *N. ausubeli* or *N. ferruginous* infection, one gene specific to *N. parisii* infection, and no genes specific to *N. ironsii* infection. We also observed

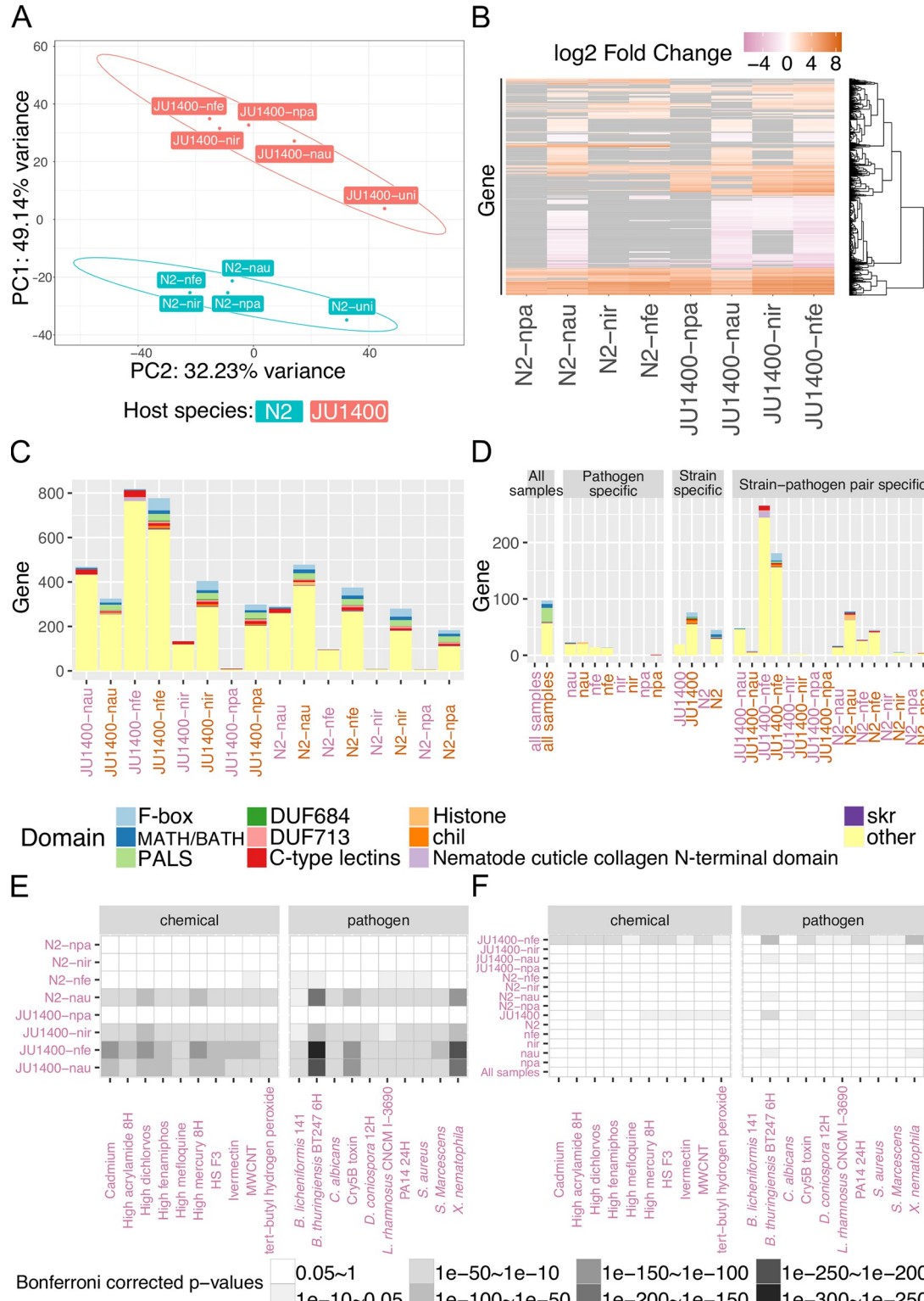

**Fig 7. RNA-seq analysis of N2 and JU1400 infected with four species of microsporidia.** N2 and JU1400 animals were infected with one of four species of microsporidia (see Methods for doses) for 48 hours and RNA was extracted and sequenced. (A) Principal component analysis of uninfected and infected N2 and JU1400 samples. Circles represent confidence ellipses around each strain at 95% confidence interval. (B) Heatmap of transcriptional profiles of differentially regulated genes in N2 and JU1400 samples. Rows represent genes clustered hierarchically. Scale is of differential regulation of infected compared to uninfected

samples. Grey tile indicates that the gene is not significantly differentially regulated in the sample. (C-D) Domain enrichment analysis of significantly upregulated and downregulated genes in each sample (C) and of specific gene sets (strain, pathogen, and strain-pathogen pair) (D). (E-F) Published datasets of differentially expressed *C. elegans* genes upon exposure to chemicals or bacterial infection were compared for overlap with the downregulated genes from each sample (E) or for overlap between differentially expressed genes in strain specific, pathogen specific, or strain-pathogen sets (F). (A-F) uni (uninfected), npa (*N. parisii*), nau (*N. ausubeli*), nir (*N. ironsii*), and nfe (*N. ferruginous)*. Upregulated samples are represented with orange text; downregulated samples are represented with pink text.

45 or more differentially upregulated genes that were specific to either N2 or JU1400. To determine the types of differentially expressed genes in these samples, we annotated them using a set of domains previously observed to be regulated during microsporidia infection [47,48,54] (**S9 Table**). About 35% of the conserved response were genes containing these domains, with the most frequent being PALS, F-box, and MATH/BATH domains. About 25% of the strain-specific genes also contained these domains, with each strain having at least one PALS, F-box, and MATH/BATH, c-type lectin, and skr domain-containing gene expressed that is specific to that strain (**Fig 7D** and **S10 Table**). Skp-related proteins (*skr*) in *C. elegans* were shown to bind to CUL-6 to ubiquitylate specific proteins [55]. *skr-4* is upregulated in both host strains infected by all microsporidia species. Among the upregulated genes, we also observe strain-specific *skr* genes such as JU1400 specific *skr-6*, and N2 specific *skr-5* (**S8 Table**).

To understand how transcriptional responses correlated with resistant and susceptible phenotypes, we analyzed genes that were specifically regulated in JU1400 when infected with either *N. ironsii* or *N. ferruginous* (**Fig 7D**). Surprisingly, we only observed three differentially expressed genes specific to *N. ironsii* infection of JU1400. In contrast, we saw 181 upregulated and 266 down regulated genes specific to *N. ferruginous* infection of JU1400. Many of the upregulated genes contained domains enriched in the IPR. The down regulated genes contained a few genes with IPR domains, as well as 23 genes with cuticle collagen domains. These proteins with collagen domains were especially striking, as we never identified more than one of these genes having altered expression in any of the other gene sets.

To determine if other stresses had similarities to the response of JU1400 to *N. ferruginous*, we compared the set of genes specifically differentially expressed in this sample to previously published *C. elegans* transcriptional datasets [56]. We identified the top ten chemical or microbial treatment data sets that had the most significant overlap with JU1400 infected by *N. ferruginous* (**S11 Table**). We then determined the overlap of these data sets to all our samples (**Fig 7E and 7F** and **S12 Table**). This revealed that genes differentially downregulated in JU1400 in response to *N. ferruginous* infection overlapped significantly with treatments of chemical poisons such as pesticides (fenamiphos and dichlorvos), organic compounds (acrylamide and tert-butyl hydrogen peroxide), and heavy metals (mercury and cadmium). All of these treatments were previously shown to inhibit development or the production of progeny in *C. elegans* [57–60]. We also observed significant overlap of the response to these chemicals with JU1400 infected with *N. ironsii*, and both strains infected with *N. ausubeli*. These three infected samples along with JU1400 infected with *N. ferruginous* also contained the largest number of significantly down regulated genes (**Figs 7C and S19**). Additionally, we observed significant overlap of these four microsporidia infected samples to pathogenic bacteria or toxins including *Staphylococcus aureus*, *Serratia marcescens*, *Bacillus thuringiensis*, *Xenorhabdus nematophila*, and *Pseudomonas aeruginosa* (**Fig 7E and 7F**). These patterns of transcriptional overlap were the most striking with differentially downregulated genes, although some overlap with upregulated genes was also observed (**S20 Fig**).

We also performed enrichment analysis of our transcriptional data sets, observing that the upregulated genes are commonly enriched for metabolic processes and signalling pathways. In

contrast, down regulated genes are enriched for proteolysis, fatty-acid related metabolic processes and drug metabolism (**S13** and **S14 Tables).** We also analyzed the 12 genes that contain genetic variants in both JU1400 and MY1 that are within the mapped chromosome I region. None of these genes are differentially regulated in JU1400 upon *N. ferruginous* infection.

## Discussion

Animal populations often have genetic diversity to bacterial, viral, and fungal pathogens, but less is known about how microsporidia infection influences host variation [5,6]. To investigate how different microsporidia species impact genetic diversity, we measured the phenotypic responses to infection in a collection of genetically diverse *C. elegans* wild isolates. A total of 13 out of 22 wild isolate strains displayed a significant difference in population fitness after microsporidia infection when compared to the population as a whole. We detected these interactions among three of the four microsporidia species we measured, with *N. ironsii* and *N. ferruginous* accounting for the bulk of the fitness interactions observed. We attributed these fitness changes as signatures of susceptibility (fitness decreases) or resistance (fitness increases) with respect to microsporidia infection.

So far, the causative gene for any host variants that influence microsporidia interactions has not been identified. Previous genetic mapping in *C. elegans*, *D. magna*, and honey bees have revealed 3–6 loci involved in resistance and sensitivity to microsporidia [19,23–26]. Additionally, some loci in *D. magna* appear to be specific to different microsporidia species, which we show to be the case for JU1400 [24]. Our genetic analysis of JU1400 identified that susceptibility to *N. ferruginous* in both JU1400 and MY1 mapped to a single region on the left arm of chromosome I while the resistance phenotype of JU1400 to *N. ironsii* appears to be caused by at least three genomic regions. Taken together, our results add to the increasing evidence that suggests that microsporidia have had a major impact on the evolution of animal genomes.

Microsporidia infect a wide diversity of animal phyla and infections are highly prevalent [12,13,61]. Nine species of microsporidia have been reported to infect *C. elegans*, with four having been found in wild isolates and five capable of infecting *C. elegans* after isolation of microsporidia from other nematode species [15,19,32,41]. The most common are the intestinal-infecting *N. parisii* and *N. ausubeli*, which have been found in Europe, the United States, India, and Africa. The geographical relationships between the strongest interactions we detected is not clear; JU1400 was isolated in Spain, *N. ironsii* in Hawaii, and *N. ferruginous* in France. Although host genetic factors were shown to be responsible for the geographic range of a microsporidian infecting *D. magna*, further work is needed to understand if there is a geographic basis to microsporidia resistance and susceptibility in *C. elegans* [62].

Pathogen tolerance is being increasingly appreciated as a valuable way for hosts to improve fitness in the presence of pathogens, although these mechanisms are less understood [63]. Tolerance is generally defined as the ability of a host to limit the consequences of a given level of infection [64]. Here we show that JU1400 is very sensitive to infection by two different epidermal infecting microsporidian species, but this strain does not display significantly higher levels of *N. ferruginous* infection than non-sensitive control strains. RNA-seq analysis reveals that JU1400 has a unique response to microsporidia when infected with *N. ferruginous*. This response shares similarities to several chemical toxins and pathogens that have been shown to damage *C. elegans*. Although the mechanistic basis of this transcriptional similarity is likely different between the various conditions, this result suggests that JU1400 infected with *N. ferruginous* has a response similar to worms being exposed to toxic conditions. This signature is also detected to a lesser extent in other strain-microsporidia pairings that cause developmental delay, including *N. ausubeli* with either strain or *N. ironsii* with

JU1400 [16]. Thus, one potential consequence of the lack of tolerance to *N. ferruginous* is a developmental delay. Consistent with this result is the observation of many genes containing collagen domains being down regulated in JU1400 infected with *N. ferruginous*. Further work uncovering the genetic variant responsible for this aberrant response will help elucidate the mechanistic basis of tolerance to microsporidia infection. None of the differentially regulated genes are among the list of 12 candidate genes in the critical region of chromosome I for *N. ferruginous* sensitivity, suggesting that causative gene or variant is not differentially regulated during infection.

Our initial observation from our PhenoMIP experiment is that JU1400 has severely reduced fitness when infected with *N. ferruginous* and a moderate reduction of fitness when infected with *N. ironsii* (**Fig 3A**). In our follow up experiments using embryo counts as a proxy for fitness, we observed that at 72 hours there was no relative difference in reduction of embryos during infection with *N. ironsii* (**Fig 4A**) and significantly less reduction of embryos in JU1400 at 96 hours (**Fig 4B**). However, we also observed a decrease in embryos at 72 hours when JU1400 was infected with *N. ironsii* compared to N2 or VC20019 (**S4B Fig**). We also performed MIP-MAP experiments using similar conditions to those in our initial PhenoMIP experiment. These experiments showed that JU1400 regions were selected for at both 72 hours (**S13B Fig**) and at 96 hours (**Fig 6B**). Regions on both chromosome II and chromosome V were selected for at both times, but only the region on the X chromosome was selected for at 96 hours. In addition to our experiments on host fitness, we also provide strong evidence that there is resistance to *N. ironsii* in JU1400 due to its ability to clear the parasite. Thus, our evidence is supportive of a model where JU1400 is resistant to *N. ironsii* infection and has increased fitness compared to control strains at a 96-hour time point. Our data is inconclusive about the impact of *N. ironsii* on JU1400 at 72 hours as we have conflicting results depending on the experiment. Our results suggest that a limitation of our PhenoMIP experiment is that resistance to microsporidia infection doesn't always result in a fitness advantage under certain conditions. Our data also suggests that future competitive selection experiments using microsporidia could be improved using 96-hour selections to identify resistant strains.

JU1400 is resistant to *N. ironsii* infection by clearing the invaded parasite. This immune mechanism is similar to previous results seen in *N. ironsii* infection of CB4856. However, the immunity in JU1400 results in less infection than in CB4856. The JU1400 resistance mechanism is specific, as the clearance we observe does not occur in even the closely related sister species *N. parisii*, which shares 92% identical DNA [43]. A strain of mosquito that is resistant to infection by a parasitic nematode was shown to rely on activation of an upregulated immune response [65]. However, we see no evidence for an increased transcriptional response of JU1400 in response to *N. ironsii*. A potential caveat to these experiments is that we infected JU1400 at the L1 stage, but measured the transcriptional response at the L4 stage, a stage where if infections are initiated, parasite clearance is not observed. We do, however, observe additional parasite clearance between 24 and 72 hours (**Fig 5A**), which corresponds to between the L2/L3 stages and adults, suggesting that if there was transcriptional signature of resistance, it would be observed under our experimental RNA-seq conditions. Instead of a specific transcriptional response of JU1400 to *N. ironsii*, we observe a *C. elegans* strain-specific response of immune genes including upregulation of several CHIL genes, which have been observed to be upregulated in response to epidermally infecting oomycetes, but have not previously been observed in response to microsporidia infection [66,67]. This strain-specific-response of *C. elegans* has also been observed with several bacterial pathogens, suggesting that there exists a diversity of innate immune responses within the *C. elegans* population [68]. We also observe more strain-specific regulated genes than pathogen-specific regulated

genes, suggesting that *C. elegans* responds similarly to infection by different species of *Nematocida* [69].

Here we demonstrate that PhenoMIP can be used to identify variants of pathogen infection in a high-throughput manner. We compressed the equivalent of 2898 individual infection experiments into 127 pooled population experiments, allowing a large savings in time and reagents required for this study. The genomic loci responsible for these phenotypes can then be identified using MIP-MAP [52]. We only tested a small fraction of the current *C. elegans* isotypes identified, suggesting there is much variation to microsporidia infection within *C. elegans*. Our pooled selection approach could also be applied to other *C. elegans*-related microbes such as those isolated from the *C. elegans* microbiome or known pathogens and viruses [70,71]. Here we studied the animal's reproductive fitness, but our approach could be modified, for instance, to select for pathogen load using fluorescently labelled pathogens. The wild isolate collections of *C. elegans* provide a unique opportunity to study how animals evolve in response to infection. The *Caenorhabditis elegans* Natural Diversity Resource (CeNDR) collection, hosts more than 1500 wild isolate strains across 550 isotypes [33]. Based on our initial results, many of these strains are likely to have fitness interactions with microsporidia infection. Identifying additional sensitive or resistant strains would drive the discovery and further elucidation of evolved mechanisms of host-responses in coevolving pathogens.

## Methods

### Nematode maintenance and worm pooling

Nematode strains were maintained at 21˚C on a modified version of nematode growth media NGM) with a 1:1 ratio of Agar:Agarose to reduce burrowing by wild isolate strains. NGM plates were seeded with OP50-1 bacteria grown to saturation overnight in lysogeny broth (LB) at 37˚C and concentrated to 10X. All wild isolate strains used (**S1 Data**) are available from the CGC. The AWR133 mapping strain was generated from VC20019 by inserting a GFP at I:2,851,135 using CRISPR-based homologous recombination [72]. Nematode strains were grown continuously without any starvation or contamination for a minimum of two weeks before seeding individual strains to two 15-cm NGM plates seeded with 1.5 ml of 40X concentrated OP50-1. After 96 hours, gravid adult populations were bleach-synchronized [73] and allowed to hatch into axenic M9 media for ~16 hours at 20˚C. Synchronized L1 samples were counted and equal numbers of animals were mixed into a single population before redistributing ~13,000 L1s to each 15-cm NGM experimental plate with a mixture of 1.5 ml of 40X OP50-1 and the appropriate amount of spores as needed (**S2 Table**).

### PhenoMIP competitive assays

PhenoMIP competitive assays were based on a previously described protocol [39]. We modified this method to use bleach-synchronization between generations to directly measure egg-production as an output of population fitness. At each generation, samples were bleach-synchronized as gravid adults ~72 hours after plating L1s and incubating at 21˚C. Bleached embryos were then allowed to hatch over 18–22 hours while incubating on a shaker at 21˚C in axenic M9 media. Each sample was gently pelleted and 3–7 μl of each pellet was used to seed the experimental plate for the next generation of animals. Overall, the competitive assay consisted of four total cycles, with one infection generation and one uninfected (resting) generation per cycle. For each infection cycle, 3 ml of 40X OP50-1 culture was mixed in 15-ml falcon tubes with twice the indicated dose of spores used on each infection experiment before splitting the mixture to each duplicate plate. Worm populations for each specific replicate were then added to the plate and samples were distributed evenly across the NGM surface before

drying. For each rest cycle, worm pellets were plated onto 15-cm NGM plates previously seeded with 1.5 ml of 40X OP50-1 culture. Prior to bleach-synchronizing the 72h-resting generation, a subsample of the population was collected and frozen as a genomic DNA source. Microsporidia doses were adjusted after the first infection round in order to ensure high enough selection pressure throughout the population without causing the populations to become completely infertile or developmentally arrested (S2 Table) [74]. The initial infection round with *N. ausubeli* caused animals on the high-dose plates to produce no progeny. Therefore, the medium-dose infection series became the high-dose series at 53,000 spores/cm$^2$. A new medium-dose series was started using an uninfected control replicate population from the first rest phase. This new medium-dose series (39,800 spores/cm$^2$) therefore only experienced three rounds of infection selection of *N. ausubeli*.

### PhenoMIP sequencing and analysis

PhenoMIP sequencing libraries were generated using previously designed MIPs for the wild isolate strains from the Million Mutation Project [39]. Individual MIPs were normalized to a concentration of 100 μM and pooled to a maximum volume of 85 μl. 10 μl of 10X Polynucleotide Kinase (PNK) Buffer and 5 μl of PNK were added to a volume of 85 μl pooled MIPs before incubating for 45 minutes at 37°C and 20 minutes at 80°C. This pool was then diluted to a working concentration of 330 nM. Libraries were generated using 160 ng of genomic DNA from each sample and generated as previously described [52]. Libraries were sequenced on an Illumina NextSeq system with each PhenoMIP library ranging between 0.0882x10$^6$ and 2.723x10$^6$ reads and a mean of 9566 ± 7994 reads per probe.

For each sequencing library, reads were initially analysed as previously described [52] with the exclusion of the normalization step for each MIP. After the abundance of each MIP was calculated, an average abundance was calculated for each strain as well as a standard deviation across this average. These values were used in downstream analysis of population structure across multiple timepoints.

Population structure and fold-change analysis was calculated across each experiment using the amalgamated data from above. Strains with a starting abundance value below 5x10$^{-3}$ (CB4854 and JU312) were eliminated from downstream population analysis. Remaining data were further transformed with any values below 1.0x10$^{-3}$ being converted to this value to accommodate log$_2$ growth analysis. Total fold-change and mean fold-change were calculated based on starting and end-point changes in abundance versus total generations (two generations per infection cycle). In samples with negative trajectories, however, the final generation of growth was calculated as the first instance of abundance at or below the lower limit of 1.0x10$^{-3}$. Mean fold-change rate was calculated based on the total fold-change abundance in the final generation of growth divided by the expected number of generations passed. Each experimental condition may have had multiple biological replicates (referred to in text as replicates), each with a time series of samples. Each fold-change rate calculated for a single time series represented a single replicate. Mean fold-change rate for each strain was therefore a combination of multiple replicates across multiple experimental conditions (S1 Data).

Given the nature of the smaller datasets and the expectation of potential outliers stretching out the tails of the mean FCR distribution, modified z-scores were calculated using the following formula where $\tilde{Y}$ is the median of the data and $MAD = median(|Y_i - \tilde{Y}|)$

$$M_i = \frac{0.6745(x_i - \tilde{x})}{MAD}$$

To identify outliers and potential interaction candidates, we used a minimum z-score of ± 1.5 as our threshold where both replicate samples were each required to meet the threshold.

## Microsporidia infection assays

Continuous microsporidia infection assays were conducted using ~1,000 bleach-synchronized L1s obtained from cultures rotating at 21˚C for 18–22 hours. Worm populations were pre-mixed in a 1.5 ml microfuge tube with 400 µl of 10X OP50-1 and microsporidia spores (**S2 Table**). Samples were then spread onto individual unseeded 6-cm NGM plates and allowed to dry before incubating at 21˚C. Plate populations were imaged using a Leica S9i Stereomicroscope before fixing samples for microscopy. Animals were washed in M9/0.1% Tween-20 at least three times before fixing with acetone. Chitin staining for microsporidia and embryos was performed using Direct Yellow 96 [75]. Meronts and sporoplasms were visualized using FISH probes conjugated to CAL Fluor Red 610 (LGC Biosearch Technologies). *N. ferruginous* was stained using 3 ng/ul of a mix of three CAL Fluor Red 610 probes designed to target the 18S rRNA: GTTGCCCGCCCTCTGCC, CTCTGTCCGTCCTCGGCAA, and CGGTCTCTAATCGTCTTC. For pathogen-specific staining, MicroF-CAL Fluor Red 610 (AGACAAATCAGTCCACGAATT) was used for *N. parisii* and Microsp1A-FAM (CAGGTCACCCCACGTGCT) was used for *N. ausubeli*. Unless otherwise noted, all remaining FISH probes used 5 ng/µl of MicroB, a *Nematocida* microsporidia 18S rRNA sequence: CTCTCGGCACTCCTTCCTG [15]. Stained samples were mixed with Biotium EverBrite mounting medium with DAPI before mounting on slides for quantitation.

## Embryo analysis and normalization

Unless otherwise noted, embryo analyses were completed using normalized values which were calculated in the following manner. For each replicate experiment, the uninfected control for each strain was used to establish a baseline mean embryo value ($\bar{Y}$). Embryo counts for individual animals ($Y$) of the same strain within the same replicate experiment were normalized ($Y/\bar{Y}$) before combining with the normalized values from matching replicate experiments. This normalization step facilitated the analysis of relative changes to embryo production, rather than absolute changes in embryo counts which could vary both between nematode strains and replicate experiments. The combined normalized populations were compared using two-way ANOVA with a Tukey HSD post hoc.

## Microscopy and image quantification

All imaging was done using an Axio Imager.M2 (Zeiss) and captured via ZEN software under identical exposure times per experiment. Animals containing clumps of spores visible in the body after 72 hours by DY96 straining were considered as 'new spore formation'. Animals with one sporoplasm or more in intestinal cells stained with FISH were considered infected. Gravidity of worms was determined by counting the number of embryos per hermaphrodite animal.

Pathogen burden and body size was determined using ImageJ/FIJI software [76]. Individual worms were outlined as "region of interest" with selection tools. Body size of worms were determined using the "measure area" function. To quantify fluorescence (pathogen burden), signal from FISH staining was subjected to the "threshold" function followed by "measure percent area" tool.

## Pulse-chase assays

Pulse-chase assays were performed on 6-cm NGM plates with roughly 6,000 bleach-synchronized L1s obtained from cultures rotating at 21˚C for 18–22 hours. Worm populations were first premixed in a 1.5 mL microcentrifuge tube with 5 µL of 10X OP50-1 and 1.25 million spores per microsporidia strain. This dose was chosen to produce N2 controls with 85–90% infection rates at 3 hpi. Samples were then spread onto individual 6-cm NGM plates and allowed to dry before incubating at 21˚C for 3 hours. To subsample the population at 3 hpi, plates were washed at least two times with M9 in a 1.5 mL microcentrifuge tube to remove OP50-1 and residual spores. Half the population was then fixed in acetone and stored at -20˚C. The other half was mixed with 50 µL of 10X OP50-1 and plated onto 6-cm NGM plates. Samples were then incubated at 21˚C for 21 hours (24 hpi), washed, subsampled, fixed, and replated (with 300 µL of 10X OP50-1) in the same manner as described above. The plates were then incubated at 21˚C for 48hours (72hpi) before washing and fixing in acetone. Pulse-chase co-infection assays were performed as described above, except samples were infected with either one or two species of microsporidia for 24 hours total. Pulse-chase assays initiated at the L4 stage were performed by using ~3000 bleach-synchronized L1s mixed with 200 µL of 10X OP50-1 and plated on unseeded 6-cm NGM plates for 48 hours. Worm populations were then washed two times with M9 in 1.5 mL microcentrifuge tubes. 50 µL of 10X OP50-1 and spores were added to each worm population and then plated on unseeded 6-cm NGM plate for 3 hours at 21˚C. Samples were then subsampled, washed, and replated as described above. Chitin and FISH staining were performed as described above for all pulse-infection experiments.

## MIP-MAP and variant analysis

Wild isolates were mapped using a competitive fitness approach after crossing with males of the mapping strain AWR133, a CRISPR-inserted Pmyo-2::GFP version of VC20019. Mapping was completed using 2–4 competitive selection cycles on 10-cm NGM plates. Each cycle consisted of 2 generations of recombinant populations, each added to plates as bleach-synchronized L1 animals. In the first generation ~10,000 L1 cross-progeny animals were exposed to continuous microsporidia infection (See **S3 Table**) for ~72 hours before bleach-synchronizing. Next, ~8,000 L1s were plated within 24 hours to "rest" on 10-cm OP50-seeded NGM plates and grown for ~72 hours before being bleach-synchronized. L1 progeny of the "rest" phase were then plated onto both an OP50-1-seeded plate with spores to begin the next cycle and a secondary plate without spores to generate a sample to collect genomic DNA. Genomic samples were prepared with 250 ng of genomic DNA and VC20019 MIP-MAP library probes as previously described [52]. Sequencing libraries were run on an Illumina NextSeq, with 161,054–801,840 reads per library and a mean of 4005 ± 2760 reads per MIP across the entire data set.

## Near isogenic lines construction

Near isogenic lines were tracked across chromosome I using four markers located at four locations matching sequenced variants identified in JU1400: cewivar00066519 (491,763), cewivar00067570 (1,051,810), cewivar00069114 (1,567,450), and cewirvar00070489 (2,012,869). These markers were used for restriction fragment length polymorphism analysis with the enzyme DraI. JU1400 hermaphrodites were crossed with AWR133 males. F1 GFP+ animals at the L4 stage were chosen to produce F2 animals. 48 GFP+ animals were chosen and screened at marker cewivar00066519 for JU1400 homozygosity and further screened at cewirvar00070489 for JU1400 homozygosity or heterozygosity. Seven F2 lines were chosen based on their genotyping and four of these gave rise to five main progenitor NILs, each of which was used to create a separate GFP+ and GFP- NIL strain. In total 10 NILs were generated and

tested for susceptibility against LUAm1. One line was chosen for outcrossing (AWR140) which gave rise to the AWR144 and AWR145 NILs.

### C. elegans whole genome sequencing analysis

Worm strains used for whole genome sequencing were grown on 10-cm plates with OP50-1 for 3 days and allowed to starve. Plates were then washed at least two times with M9 in micro-centrifuge tubes before freezing in liquid freezing solution and then stored at -80C. gDNA was then extracted and quantified with Qubit HS system. gDNA quality was determined by agarose gel electrophoresis and photometric OD260/280 of 1.8–2.0. gDNA was submitted to The Centre for Applied Genomics (TCAG, Hospital for Sick Children, Toronto, ON, Canada) for library preparation and whole genome sequencing on an Illumina HiSeq 2500 SR50 Rapid Run flowcell in pools of six samples per lane. Raw sequencing reads of each NIL were aligned to the N2 reference of the WBcel215 release (accession: PRJNA13758) using bowtie2 [77]. Then, samtools was used to convert it into.bam format and freebayes was used to perform variant calling with the default settings [78,79]. The breakpoints between VC20019 and JU1400 in chromosome I of each NIL were computed and visualised using vcf-kit [80].

### RNA extraction and mRNA sequencing

Infection samples for mRNA sequencing were prepared by mixing ~10,000 bleach-synchronized L1 stage animals with 1 mL of 20X OP50-1 and 10 million *N. parisii* spores, 10 million *N. ausubeli* spores, 10 million *N. ironsii* spores, and 30 million *N. ferruginous* spores in four separate microcentrifuge tubes. Samples were then spread onto individual 10-cm NGM plates and allowed to dry before incubating at 21˚C for 48 hours. Samples for RNA extraction were washed in M9/0.1% Tween-20 in 15 mL conical tubes at least three times before adding 1 mL of TRI-Reagent (Sigma-Aldrich). Total RNA extraction and ethanol precipitation protocols was performed as per Tri-Reagent manufacturers instruction. Libraries were prepared using NEBNext Ultra™ II Directional RNA Library Prep Kit for Illumina kit as per manufacturer's instruction. RNA sequencing was carried out using Illumina NovaSeq-SP flow cell by The Centre for Applied Genomics (TCAG).

### Microsporidia RNA-seq analysis

Using Bowtie2 v2.4.4, indexes were built from *Nematocida parisii* strain ERTm1 (Genbank: GCF_000250985.1), *Nematocida ausubeli* strain ERTm2 (Genbank: GCA_000738915.1), *Nematocida parisii* strain ERTm5 (Genbank: GCA_001642415.1) and *Nematocida ferruginous* strain LUAm1 (Accession: JALPMW000000000) The genome annotation file of each *Nematocida* species was converted into gtf format using Gffread v0.12.3. Paired-end reads were aligned to corresponding index and genome annotation files using TopHat v2.1.2. Cufflinks v2.2.1 was used to generate a transcriptome assembly for each sample, which was then combined with the corresponding reference genome annotation using Cuffmerge of the Cufflinks 2.2.1 package to produce a single annotation file.

### C. elegans mRNA-seq analysis

The paired end reads generated for each strain were submitted as separate projects to Alaska v1.7.2 (http://alaska.caltech.edu), which is an online automated RNA-seq analysis tool in collaboration with Wormbase. Briefly, Bowtie2 [77], Samtools [78], RseQC [81], FastQC, and MultiQC [82] were used for quality control of the input files. Kallisto [83] was used for the read alignment

and quantification, followed by differential analysis by Sleuth. For both N2 and JU1400 samples, the reads were aligned to the N2 reference of the WS268 release (accession: PRJNA13758).

### Principal component analysis

The N2 and JU1400 normalized abundance measurements generated by Alaska, were read by the readRDS() function of the R package base v4.1.1. Normalized counts for each gene in each strain were generated via sleuth_to_matrix() with the "obs_norm" data and "tpm" units. PCA of gene expression was performed using samples from each strain, and merged samples, via the R package pcaexplorer v2.20.2 [84].

### Differentially expressed genes of N2 and JU1400 hosts

$Log_2$ fold change values of duplicated genes in each of the eight samples were averaged using the R package dplyr v1.0.8. Genes with FDR-adjusted p-value of <0.01 were regarded as significant. Significant genes expressed in at least three out of the eight samples were used for hierarchical clustering via the function hclust() from the R package stats v4.1.1, then plotted using R packages ggdendrogram and ggplot2 v3.3.5. Differentially upregulated genes were defined as those with an FDR-adjusted p-value<0.01 and $log_2$ fold change >0 (infected vs. control); differentially downregulated genes were defined as those with an FDR-adjusted p-value <0.01 and $log_2$ fold change <0 (infected vs. control).

### Strain-specific, pathogen-specific and strain-pathogen pair specific genes

Strain-specific genes were defined as genes that are significantly upregulated or downregulated in at least three out of the four JU1400 or N2 samples. Pathogen-specific genes were defined as those significantly upregulated or downregulated in both JU1400 and N2 upon infection by the same microsporidia species, but not observed when infected by other microsporidia species. Moreover, strain-pathogen pair specific genes were defined as genes that are significantly upregulated or downregulated in a particular strain-pathogen sample, but not observed in the other seven strain-pathogen samples.

### Domain enrichment analysis

We chose to look at F-box, MATH/BATH, PALS, DUF684 and DUF713 domains containing genes, implicated from differentially regulated genes of *N. parisii* infected *C. elegans* in previous publications [54,85]. Lists of gene names from respective gene classes were downloaded separately from Wormbase (http://wormbase.org/) and the corresponding protein-coding sequences were extracted from Wormbase ParaSite (https://parasite.wormbase.org/). Alignments of other domains or classes were downloaded directly from Pfam (http://pfam.xfam.org/) separately. After deleting duplicated sequences, protein sequences of these classes of genes were converted to Stockholm format using Clustal Omega Multiple Sequence Alignment tool (https://www.ebi.ac.uk/Tools/msa/clustalo/). Hmmbuild of HMMER v3.3.2 [86] used the Stockholm format files to build respective profile HMMs, which was then searched against *C. elegans* (accession: PRJNA13758) proteome of WS268 release using hmmsearch. Output genes with E-value <1e-5 were used as the gene list for the enrichment analyses. In the output genes, MATH and BATH genes were combined into one list; fbxa, fbxb and fbxc genes were merged into the gene list for F-box (**S10 Table**). Using the R package gplots v3.1.1, the number of genes overlapping between the lists and samples were computed, then plotted using R package ggplot2 v3.3.5. Genes that do not fall into any of these gene classes or have any of these domains were categorised as "other".

## Gene set enrichment analysis

The WormExp [56] dataset, which contains published expression data for *C. elegans*, was downloaded from the WormExp v1.0 website (https://wormexp.zoologie.uni-kiel.de/wormexp/). The overlap between published datasets of *C. elegans* exposed to different chemicals or pathogens (**S12 Table**) and our samples, as well as the strain specific, pathogen-specific, and strain-pathogen pair specific genes in our dataset were computed using the R package gplots v3.1.1. For the statistical evaluation of the overlap, the p-values were calculated using the adjusted Fisher exact test method from the program EASE, which removes one gene in the set of intersected genes. Correction for multiple testing was implemented using the Bonferroni method via the p.adjust() function in the R package stats v4.1.1.

## GO biological process, GO molecular function and KEGG pathway enrichment analyses

Using the R package enrichR v3.0 [87], the connection to WormEnrichr, a gene list enrichment analysis tool for *C. elegans*, was set through the setEnrichrSite ("WormEnrichr") function. The "GO_Molecular_Function_2018", "GO_Biological_Process_2018" and "KEGG_2019" databases were selected then used to query enrichr with the enrichr() function, before being plotted as Enrichr GO-BP output **(S13 and S14 Tables)**.

## Volcano plots

Values of duplicated genes in each of the four host-microsporidia samples were averaged using the R package dplyr v1.0.8. Genes with FDR<0.05 and $\log_2$ Fold Change >0 or $\log_2$ Fold Changes <0 were categorised as differentially upregulated or downregulated respectively. The volcano plots were generated by ggplot2 v3.3.5 in R.

## Supporting information

**S1 Fig. Geographical locations of *C. elegans* and microsporidia used in this study.** (A) World map highlighting the locations where the *C. elegans* wild isolate strains (orange circles), *Nematocida* species (green triangles), and the VC20019 reference strain (purple star) used in this study were originally isolated. Map of the world is from ggplot2 version 3.3.5. https://rdrr.io/cran/maps/man/world.html.
(TIF)

**S2 Fig. Analysis of PhenoMIP mean fold-change rate identifies candidate nematode/microsporidia interactions.** Violin plots of 20 wild isolates and lab reference-derived control strain VC20019 showing modified z-scores of mean fold-change rate (FCR) per generation. Violin shapes are derived from the eight mock-infection replicates. Overlayed are the eight mock infections replicates and six infection replicates for each of *N. parisii* (A), *N. ausubeli* (B), *N. ironsii* (C), and *N. ferruginous* (D). Replicates are coloured by mock infection (yellow), low dose (green), medium dose (blue), and high dose (purple) with increasing point size associated with increasing dose. Dotted lines represent the z-score candidate threshold of ±1.5. The strains CB4854 and JU312 were excluded from analysis due to low initial population abundance.
(TIF)

**S3 Fig. The Hawaiian (CB4856) strain shows resistance to infection by high-dose *N. ironsii*.** Line graph of fold change across multiple time points and conditions for CB4856 demonstrates potential fitness increase relative to mock-infection conditions when infected with a high dose

of *N. ironsii*. Microsporidia strains are denoted by point shape and doses of microsporidia are denoted by line type. The eight uninfected control replicates are bounded within the greyed area with the mean of the uninfected controls denoted by a solid red line. All other lines represent mean FCR of two biological replicate samples.
(TIF)

**S4 Fig. Single-strain testing of PhenoMIP interaction candidates with *N. ferruginous* infection.** (A-C) Strains of *C. elegans* were infected with medium and high doses of *N. ferruginous* for 72 hours, imaged, fixed, and stained with DY96 and a *N. ferruginous* 18S rRNA FISH probe. (A) images of N2 and VC20019 controls versus AB1, JU397, ED3042, and JU360 at 72 hpi. (B) Dot plot with mean and SEM bars of normalized embryo counts from infected populations with n = {41,70} worms per sample. (C) Bar plot depicting the percent of animals with detectable meront staining. p-values denote comparison to N2 controls and were determined by two-way ANOVA with Tukey post-hoc. Significance was defined as $p \leq 0.001$ (***) and not significant as $p > 0.05$ (ns).
(TIF)

**S5 Fig. Single-strain testing of PhenoMIP interaction candidates with *N. ironsii* infection.** (A-C) Strains of *C. elegans* were infected with medium and high doses of *N. ironsii* for 72 hours, imaged, fixed, and stained with DY96. (A) images of N2 and VC20019 controls versus CB4856, JU300, JU1400, and MY2 at 72 hpi. (B) Dot plot with mean and SEM bars of normalized embryo counts from infected populations with n = {45,115} worms per sample. (C) Bar plot depicting the percent of animals with newly generated spores. p-values denote comparison to N2 controls and were determined by two-way ANOVA with Tukey post-hoc. Significance was defined as $p \leq 0.001$ (***), $p \leq 0.01$ (**), $p \leq 0.05$ (*) and not significant as $p > 0.05$ (ns).
(TIF)

**S6 Fig. Single-strain testing of PhenoMIP interaction candidates with *N. ausubeli* infection.** (A-C) Strains of *C. elegans* were infected with medium and high doses of *N. ausubeli* for 72 hours, imaged, fixed, and stained with DY96. (A) images of N2 and VC20019 controls versus JU360 and MY2 at 72 hpi. (B) Dot plot with mean and SEM bars of normalized embryo counts from infected populations with n = {54,71} worms per sample. (C) Bar plot depicting the percent of animals with newly generated spores. p-values denote comparison to N2 controls and were determined by two-way ANOVA with Tukey post-hoc. Significance was defined as $p \leq 0.05$ (*) and not significant as $p > 0.05$ (ns).
(TIF)

**S7 Fig. JU1400 and MY1 share similar response phenotypes to infection by *N. ferruginous*.** JU1400, MY1, and N2 L1s were infected with a medium dose of *N. ferruginous* for 72 hours, fixed, and stained with *N. ferruginous* 18S rRNA FISH probes and DY96. (A) Dot plot with mean and SEM bars of embryo counts from infected populations. (B) Bar plot with SEM bars depicting the percent of animals with visible meronts by FISH staining. Data is combined from three biological replicates with n = {41,50} worms per replicate sample. p-values denote comparison to N2 controls and were determined by two-way ANOVA with Tukey post-hoc. Significance was defined as $p \leq 0.01$ (**), $p \leq 0.05$ (*) and not significant as $p > 0.05$ (ns).
(TIF)

**S8 Fig. JU1400 and N2 display similar levels of *N. ferruginous* infection.** (A-D) JU1400 and N2 L1s were infected with 1.25 million spores of *N. ferruginous* for 3 h, washed to remove spores, and replated for an additional 69 h. Animals were fixed at 3 hpi and 72 hpi and stained

with *N. ferruginous* 18S rRNA FISH probes. (A) Representative images of infected animals at 72 hpi. Scale bars are 100 μm. (B) Bar plot with SEM bars depicting the percent of animals displaying either sporoplasm (3 hpi) or meronts (72 hpi) by FISH staining. (C-D) quantitation of FISH stain at 72 hpi. (C) Dot plot with mean and SEM bars of parasite burden by area in infected populations with a minimum of n = {41,50} worms per sample. (D) Bar plot with SEM bars depicting mean parasite burden as a function of percent fluorescent area per animal by FISH staining. Data in (C, D) is combined from three biological replicates with n = 40 worms per replicate sample. p-values denote comparison to N2 or JU1400 controls and were determined by two-way ANOVA with Tukey post-hoc. Not significant was defined as $p > 0.05$ (ns).
(TIF)

**S9 Fig. JU1400 displays a similar response to *N. ferruginous* strains and sensitivity to *N. cider*.** (A-D) N2 and JU1400 L1 stage animals were infected with medium doses of *N. ferruginous* strain LUAm1 or LUAm3 (A-B) or a medium dose of *N. cider* (C-D) for 72 hours, fixed, and stained with DY96 and 18S rRNA FISH probe. (A, C) Dot plot of normalized embryo counts of animals with mean and SEM bars. (B, D) Bar plot with SEM bars of percent population with meronts visible. Data are combined from three biological replicates with n = 50 worms per replicate. p-values were determined by two-way ANOVA with Tukey post-hoc. Significance was defined as $p \leq 0.05$ (*), $p \leq 0.001$ (***), and not significant as $p > 0.05$ (ns).
(TIF)

**S10 Fig. JU1400 is resistant to infection by *N. ironsii*.** N2 and JU1400 L1 stage animals were infected with mock, medium, and high doses of *N. ironsii* for 72 (A and C) and 96 (B and D) hours and fixed. Embryos and mature spores are visualized by staining with DY96 (green), meronts are stained by a 18S rRNA FISH probe. (A-B) Representative images of high dose infection from each experiment. Scale bars are 100 μm. (C-D) Bar plots with SEM bars of percent population with meronts visible. Data in (C, D) is combined from three biological replicates each with n = {47,50} worms per replicate. (E-F) N2, JU1400, and CB4856 animals at the L1 stage were infected for 3 hours with 1.25 million spores of either *N. parisii*, *N. ausubeli*, or *N. ironsii* before washing away excess spores. Animals were either fixed at 3 hours (E) or replated for an additional 21 hours (F) and then fixed. Shown are bar plots with SEM bars of percent population with meronts visible. Data in (E, F) are combined from three biological replicates each with n = {59,179} worms per replicate. p-values were determined by two-way ANOVA with Tukey post-hoc. Significance was defined as $p \leq 0.05$ (*), $p \leq 0.001$ (***), and not significant as $p > 0.05$ (ns).
(TIF)

**S11 Fig. JU1400, but not N2, can specifically eliminate *N. ironsii* infection.** N2 and JU1400 animals at the L1 stage were pulse infected for 3 hours with *N. parisii*, *N. ausubeli*, *N. ironsii*, *N. parisii* + *N. ausubeli*, or *N. ausubeli* + *N. ironsii* before washing away excess spores. Animals were either fixed immediately at 3 hpi or at 24 hpi. Samples were then stained with species-specific 18S RNA FISH probes (*N. ausubeli* = green; *N. parisii* and *N. ironsii* = magenta) and animals containing either sporoplasms or meronts were counted as infected. (A) Representative images of N2 and JU1400 infected with *N. parisii* and *N. ausubeli* (A) and *N. ironsii* and *N. ausubeli* (B). Scale bar 10 μm for 3 hpi and 25 μm for 24 hpi. Insets at top right for each image show an enlarged image of the outlined boxes. (C) Bar plot with SEM bars show percent of N2 animals infected per replicate from three biological replicates with n = {59, 179} worms per replicate. p-values were determined by two-way ANOVA with Tukey post-hoc. Non-significance was defined as $p > 0.05$ (ns).
(TIF)

**S12 Fig. Genetic mapping of CB4856 resistance to *N. ironsii*.** Genetic mapping of CB4856 resistance to *N. ironsii* was performed with MIP-MAP using the AWR133 mapping strain. Green square-points and dotted lines represent a mock-infection replicate with the solid blue and orange lines representing two biological replicates exposed to high dose infection conditions for 72-hours per selection cycle. A red arrow on chromosome V indicates a strong signal on the right arm. Additional red arrows on chromosomes II and IV indicate broad regions of interest across the chromosomes that may contribute to CB4856 resistance phenotypes. Golden arrow indicates the z*eel-1/peel-1* incompatibility locus at chromosome I. (TIF)

**S13 Fig. Genetic mapping of interaction phenotypes of JU1400 to *N. ironsii* and MY1 to *N. ferruginous* using 72-hour selections.** Genetic mapping of (A) MY1 susceptibility to *N. ferruginous* and (B) JU1400 resistance to *N. ironsii* was performed with MIP-MAP using the AWR133 mapping strain. Green square-points and dotted lines represent a mock-infection replicate with the solid blue and orange lines representing two biological replicates exposed to specific infection conditions. Red arrows indicate candidate regions of interest based on the expected genome fixation direction during phenotype selection. Golden arrow indicates the *zeel-1/peel-1* incompatibility locus at chromosome I. (TIF)

**S14 Fig. JU1400 and MY1 susceptibility to *N. ferruginous* is recessive and MY1 is less resistant than JU1400 to *N. ironsii*.** (A-B) Analysis of F1 progeny from crosses between AWR133 (VC20019 with integrated pmyo-2::GFP) and the wild isolates JU1400 or MY1 as well as F1 progeny from crosses between JU1400 hermaphrodites and MY1 males. F1s were examined 72 hours post-infection by *N. ferruginous* for body size (A) and normalized embryo counts (B). (B) Dot plot with mean and SEM bars of normalized embryo counts from infected populations with a minimum of n = {47,51} worms per sample. (C-D) L1 stage animals were infected for 72 hours with *N. ironsii*, fixed, and stained with DY96 and an 18S rRNA FISH probe. Bar plots depict the percent of animals with newly generated spores (C) or meront signal (D). p-values were determined by two-way ANOVA with Tukey post-hoc. Significance was defined as $p \leq 0.001$ (***), $p \leq 0.05$ (*) and not significant as $p > 0.05$ (ns). (TIF)

**S15 Fig. Generation of NILs carrying the chromosome I region that is responsible for susceptibility to *N. ferruginous*.** A schematic (A) of 10 separate NILs generated through a cross between AWR133 and JU1400. These lines were infected with a medium dose of *N. ferruginous* for 72 hours (B) and their resulting body size phenotype was assessed in comparison to N2 and JU1400 control strains. (TIF)

**S16 Fig. Microsporidia mRNA reads per sample demonstrates JU1400 resistance to *N. ironsii*.** Scatterplot of overall read mapping rate of *Nematocida* in both strains with mean and standard deviation. uni (uninfected), npa (*N. parisii)*, nau (*N. ausubeli)*, nir (*N. ironsii)*, and nfe (*N. ferruginous)*. (TIF)

**S17 Fig. Principal component analysis of N2 and JU1400 samples.** Principal component plots of expression data from only (A) N2 samples or (B) JU1400 samples. Circles represent confidence ellipses around each strain at 95% confidence interval. uni (uninfected), npa (*N. parisii*), nau (*N. ausubeli*), nir (*N. ironsii*), and nfe (*N. ferruginous)*. (TIF)

**S18 Fig. Comparison of correlation between N2 and JU1400 strains after infection with different microsporidia species.** Correlation between $\log_2$ fold changes in genes of N2 and JU1400 animals infected with (A) *N. parisii* (B) *N. ausubeli* (C) *N. ironsii* (D) *N. ferruginous*. Pearson's correlation value and p-value for each cluster (pink, green, blue, purple) are presented in the upper left corner. Pearson's correlation value and p-value for all the genes are presented in the bottom left corner. A gene with FDR adjusted p-value of <0.01 is deemed significant in each strain.
(TIF)

**S19 Fig. Volcano plots of differentially regulated genes.** (A) *N. parisii*-infected N2. (B) *N. parisii*-infected JU1400. (C) *N. ausubeli*-infected N2. (D) *N. ausubeli*-infected JU1400. (E) *N. ironsii*-infected N2. (F) *N. ironsii*-infected JU1400. (G) *N. ferruginous*-infected N2. (H) *N. ferruginous*-infected JU1400. Each point represents a gene, orange points indicate differentially upregulated genes, pink points are differentially downregulated genes while grey points represent genes with FDR-adjusted p-value >0.01 (red horizontal line). The top five differentially upregulated and downregulated genes with FDR-adjusted p-value <0.01 are labelled with their respective gene names.
(TIF)

**S20 Fig. Overlap between the significant upregulated genes upon microsporidia infection and differentially expressed datasets of *C. elegans* genes upon exposure to chemicals or bacterial infection.** (A) upregulated genes from microsporidia-infected samples. (B) upregulated gens from strain specific, pathogen specific, and strain-pathogen specific gene sets. uni (uninfected), npa (*N. parisii)*, nau (*N. ausubeli*), nir (*N. ironsii*), and nfe (*N. ferruginous)*.
(TIF)

**S1 Table. Microsporidia species information.**
(DOCX)

**S2 Table. PhenoMIP microsporidia infection doses.**
(DOCX)

**S3 Table. Wild isolate mapping microsporidia infection doses.**
(DOCX)

**S4 Table. NIL strain chromosome I region composition.**
(DOCX)

**S5 Table. Candidate genes found in chromosome I region for *N. ferruginous* sensitivity.**
(CSV)

**S6 Table. Microsporidia gene FPKM values in N2 and JU1400.**
(XLSX)

**S7 Table. Normalized *C. elegans* gene counts for all RNA-seq replicates from this study.**
(XLSX)

**S8 Table. Results of *C. elegans* differential expression analyses for all RNA-seq comparisons performed in this study.**
(XLSX)

**S9 Table. Gene classes and domains used for enrichment analyses.**
(DOCX)

**S10 Table. The number of differentially regulated genes that contain enriched domains.**
(XLSX)

**S11 Table. Published data sets from WormExp used in this study.**
(DOCX)

**S12 Table. List of gene set overlaps from this study to published data sets.**
(XLSX)

**S13 Table. GO Biological Processes, GO Molecular Functions, and KEGG Pathways enriched in N2 genes upregulated or downregulated upon microsporidia infection.**
(XLSX)

**S14 Table. GO Biological Processes, GO Molecular Functions, and KEGG Pathways enriched in JU1400 genes upregulated or downregulated upon microsporidia infection.**
(XLSX)

**S1 Data. Data of all infection experiments.**
(XLSX)

## Acknowledgments

We thank Hala Tamim El Jarkass and Keir Balla for providing helpful comments on the manuscript. We thank John Calarco for the kind gift of plasmids to perform CRISPR and Robert Waterston and Donald Moerman for the kind gift of reagents to perform PhenoMIP and MIP-MAP. Some strains were provided by the CGC, which is funded by NIH Office of Research Infrastructure Programs (P40 OD010440) and we thank WormBase.

## Author Contributions

**Conceptualization:** Calvin Mok, Aaron W. Reinke.

**Formal analysis:** Calvin Mok, Yin C. Wan.

**Funding acquisition:** Aaron W. Reinke.

**Investigation:** Calvin Mok, Meng A. Xiao, Yin C. Wan, Winnie Zhao, Shanzeh M. Ahmed.

**Methodology:** Calvin Mok.

**Resources:** Calvin Mok, Robert J. Luallen, Aaron W. Reinke.

**Supervision:** Aaron W. Reinke.

**Validation:** Calvin Mok, Meng A. Xiao.

**Visualization:** Calvin Mok, Yin C. Wan.

**Writing – original draft:** Calvin Mok, Meng A. Xiao, Yin C. Wan, Aaron W. Reinke.

**Writing – review & editing:** Calvin Mok, Meng A. Xiao, Yin C. Wan, Winnie Zhao, Shanzeh M. Ahmed, Robert J. Luallen, Aaron W. Reinke.

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
