## [Decision Letter · Decision Letter 0]

20 Nov 2022

Dear %TITLE% Reinke,

Thank you very much for submitting your manuscript "High-throughput phenotyping of infection by diverse microsporidia species reveals a wild C. elegans strain with opposing resistance and susceptibility traits" for consideration at PLOS Pathogens. As with all papers reviewed by the journal, your manuscript was reviewed by members of the editorial board and by several independent reviewers. In light of the reviews (below this email), we would like to invite the resubmission of a significantly-revised version that takes into account the reviewers' comments.

We cannot make any decision about publication until we have seen the revised manuscript and your response to the reviewers' comments. Your revised manuscript is also likely to be sent to reviewers for further evaluation.

Sincerely,

Erik C. Andersen, Ph.D.

Guest Editor

PLOS Pathogens

P'ng Loke

Section Editor

PLOS Pathogens

Kasturi Haldar

Editor-in-Chief

PLOS Pathogens

orcid.org/0000-0001-5065-158X

Michael Malim

Editor-in-Chief

PLOS Pathogens

orcid.org/0000-0002-7699-2064

Reviewer's Responses to Questions

**Part I - Summary**

Reviewer #1: In this manuscript, Mok et al perform a large-scale survey of how c. Elegans fitness is affected by diverse microsporidia pathogens. The large-scale survey identified multiple leads, of which the authors further characterized two elegans-microsporidia interactions. Overall, the main conclusions are well supported by the presented data, though there are several places where additional clarification would be helpful to readers.

Reviewer #2: The manuscript by Mok et al., describes high-throughput phenotyping of C. elegans wild isolates and their interactions with various species of microsporidia from the genus Nematocida using multiplexed competition assays. While several potential new interactions were identified, the most intriguing finding is the opposing phenotype of the isolate JU1400, which is sensitive to epidermis-infecting N. ferruginous, but resistant to intestine-infecting N. ironsii. Genetic mapping indicates that these phenotypes are likely to be driven by separate alleles. Additionally, the authors present a transcriptional analysis showing that transcriptional response to microsporidia species is largely conserved and similar to IPR, with minor strain-specific differences, but no significant correlation with the resistant or susceptibility phenotypes. Overall, it’s an ambitious attempt to understand how pathogens can shape host evolution, and this approach may be used in the future for the high throughput study of other C. elegans isolates interactions with their natural pathogens. The presentation of the data and the conclusions drawn can be improved as described with more specific comments below.

Reviewer #3: In this manuscript by Mok et al, the authors report a high-throughput phenotyping of infection by diverse microsporidia species among wild isolates and identified one strain, JU1400 with two opposing resistance and susceptibility traits. They further conducted genetic mapping and revealed multiple loci responsible for the interactions JU1400 with microsporidia. Lastly, they conducted RNAseq to examine the transcriptomes of JU1400 in response to microsporidia infection.

**Part II – Major Issues: Key Experiments Required for Acceptance**

Reviewer #1: None to note

Reviewer #2: 1. An obvious limitation is that the authors narrow down the genetic interval for one interaction to 41 protein coding changes (is there a list included in the supplement?), but present no attempt to identify the causative genes. Did the authors try to test by RNAi the best candidates in the interval on chromosome I and see whether they modulate susceptibility of JU1400 towards N. ferruginous. Alternatively, could the authors put together a hypothetical model based on known/predicted functions of these genes.

2. It has been shown that activation of IPR in the hypodermis is sufficient to provide protection from intestine infecting N. parisii(https://journals.plos.org/plosgenetics/article?id=10.1371/journal.pgen.1010314). Considering that JU1400 displays different sensitivity towards intestine and hypodermis infecting microsporidia, what happens if JU1400 is exposed simultaneously or sequentially to both N. ironsii and N. ferruginous? This would be a nice addition to the co-infection experiments performed in this study.

3. In Fig.3 E,F a single early timepoint is presented, but later time points are also required to further support the proposed susceptibility to N. ferruginous infection due to a weakened tolerance (in light of other results presented in the paper too).

Reviewer #3: This manuscript is very data-heavy. The authors have spent huge amount of efforts in getting all these data. However, the presentation of the results is often not very clear. I suggest that the authors need to revise their figures so that they can be easily followed and understood by the readers. For examples, in the text, the authors said that they identified 13 strains with phenotypes (line 129). However, it is not clear in Fig. 2, which are the 13 strains?

In Fig. 3A and B, it is very difficult to distinguish all the different lines. In all the Bar charts, there are too many lines in the background. I suggest removing them and make the charts cleaner. For the susceptibility mapping, the authors mapped to ~700 Kb region on chromosome 1. The authors only described the number of the genes in this region but did not tell us what genes they are? (line 258-259). It will be nice to at least discuss the potential genes within this locus that might be crucial for the susceptibility.

For the RNAseq section, the authors mainly focused on the number of the DEGs in each of the conditions, which is not very informative. I suggest that the authors providing a summary of GO term enrichment analyses for the different samples and draw some conclusions. I also feel that the authors should identify a couple interesting targets from their RNAseq data and follow up with functional tests. Currently the RNAseq section is lack of focus. Readers do not get much information out of these huge datasets.

**Part III – Minor Issues: Editorial and Data Presentation Modifications**

Reviewer #1: Initial experiment to assess wild isolate responses to diverse microsporidia:

The authors looked at the fitness of 22 wild-isolates and the lab strain VC20019 after exposure to four microsporidia species at four different spore concentrations. Pooled L1s were exposed to spores for 72h and bleach synchronized, allowed one generation to recover, and infected again. This two generation cycle was repeated four times for all conditions except N. ausubeli medium dose. The authors maintained worms in starvation-free conditions for two weeks prior to starting the infection experiments. Prior to infection experiments, worms were bleach synchronized and arrested in a starvation state, which the worms went through during each generation of the infection experiment. Does larval arrest after bleach synchronization have an effect on microsporidia infection? Does this vary across wild isolates? I’m mostly curious because the authors paid close attention to preventing starvation for two weeks prior to starting this assay only to expose them to starvation for the immediate generation prior to the infection experiment and all subsequent generations. I understand that this might be the only way to properly initiate infection experiments, but am curious as to the author's rationale for preventing starvation prior to inducing starvation conditions.

The authors identified several strain specific interactions from their large initial experiment to further characterize. It should be noted, and i mention this below, that of the two interactions the authors followed up, one interaction was found to be the opposite of the original observation. Though this does not affect the conclusions of the manuscript, I think there should be more discussion about this point. Despite the author’s claim on lines 390-391, I was not convinced that PhenoMIP is high-throughput approach to identify variants associated with pathogen susceptibility.

N. ferruginous section for ju1400:

The authors performed a 3hr infection pulse to determine if there were measurable differences between N2 and Ju1400, there were none, so they concluded JU1400 had a tolerance issue. Tolerance seems like a broad and unspecific term that the authors tried to explore with transcriptional profiling, which, as discussed below, did not add much clarity to the author’s experimental observations.

The authors used MIP-MAP to determine the genomic region associated with the increased sensitivity of JU1400 to N. Ferruginous and N. Ironsii. The different pathogens are associated with distinct genomic loci and the authors validate the major N. Ferruginous susceptibility locus on chromosome I by constructing NILs. I am curious if any of the differentially expressed genes in response to this pathogen map to this region of the genome and might be able to inform the causative allele? This is something that I thought was missing from the transcriptional analysis.

Later in the manuscript the authors describe transcriptional profiling of infected JU1400 exposed to N. ferruginous for 48h. Lines 308-310 the authors point to a striking observation of differential expressed genes with collagen domains without elaborating on why it was striking. I imagine one reason might be that N. ferruginous infects muscle and epidermal cells? Could another reason be that infected animals are at a different developmental stage and express a different complement of collagen-containing proteins? Overall the presentation of the transcriptional profiling experiments was quite confusing and did little to support the claim that JU1400 is less tolerant to N. ferruginous infection than N2/VC20019/AWR133, which was made evident by the infection fitness assay.

N. Ironsii and JU1400:

In the initial experiment, Ju1400 was identified to be more susceptible to N. Ironsii. Upon further investigation, the opposite turned out to be true. I think it would help readers to underscore this phenotypic switching in the discussion instead of glossing over it on lines 345-346. From the data presented, it is unclear why JU1400 would appear to be sensitive to N. Ironsii in their large assay, given it has the same number of embryos as N2 72hpi, while VC20019 does not show the same trend in the assay (figure 2). It seems that the better control for Figure 4A-B would have been VC20019 and not N2, with the unsaid understanding that N2 and VC20019 should behave similarly, despite them being different strains.

As noted above the author’s due diligence following up the JU1400 N. Ironsii interaction revealed that JU1400 is resistant to this pathogen, as measured by lower pathogen load after a brief infection and embryo counts at 96hpi. The specificity that Ju1400 has for clearing n. Ironsii is an interesting result that lays the foundation for future studies to explore the molecular details of this interaction.

While I appreciate the authors attempted transcriptional profiling to determine why JU1400 was more tolerant to N. ferruginous, as presented, it was unclear what these experiments added to this manuscript. Nevertheless, these data will likely help inform future investigations into these host-pathogen responses.

Figure 2: I appreciate the concise summary of a lot of experimental data. I do think this figure can be improved by also showing the strain trajectories for the two microsporidia species that the authors follow up in the rest of the manuscript. Perhaps highlighting the focal C. Elegans strains JU1400 in each condition, as it is the exemplary strain showing multiple species-specific interactions.

Figure 3B seems to suggest that MY1 is more fit in the presence of high doses of N. ausubeli (solid thick orange lines) than in mock infection conditions. Do the authors have any idea why this might be the case or am I interpreting this figure incorrectly?

Figure 3D, 4A-B, 6D, : it is unclear exactly how the authors arrived at embryo counts per worm and to what these counts were normalized. I am assuming these counts correspond to embryos within gravid animals, but I am just guessing.

Figure 3 A/B could look a bit cleaner by just including the microsporidia strains of interest

Reviewer #2: 1. In terms of the interactions observed for microsporidia with various wild isolates, can the authors comment whether resistant and susceptible nematode strains genetically cluster together? Also, are there any geographical region parameters (like temperature, humidity, elevation etc) that might shape the interaction observed between the microsporidia species and C. elegans wild isolates?

2. In figure 3A-B and figure S2, why two lines are represented for each condition? Are these two different replicates? I would recommend that the authors show averaged or most representative line graph in this case, as it is difficult to read this graph due to crowding of multiple lines.

3. How can the authors account for the fitness disadvantage of JU1400 in the competition assay upon N. ironsii infection? Can the authors rule out technical limitations such as variability in growth and brood size in the conditions used for the experiment rather than short-term/long-term effects of infection? Accordingly, table 1 can be a bit misleading, for example, it mentions JU1400 as susceptible to both N. ironsii and N. ferruginous, so it might be good to clarify this on this table (ie add an asterisk/footnote or something).

4. The presentation of the transcriptional analysis of JU1400 exposed to various nematocidal species can be improved. I felt that clarity is lacking on what are the take home messages in the context of the main question ie why JU1400 shows opposing phenotypes. The comparison to drugs is highly speculative and can be moved to the supplement. Additionally, this analysis might fit more before the section on genetic mapping, as it seems to give to the paper an abrupt ending otherwise and does not link to the mapping at all. The scale and colours in Figure 7C and 7D does not allow for clear visualisation of genes that are mentioned in the key.

5. The choice of VC20019 (from the mullion mutation project) as “reference” warrants justification. This is a mutagenized strain so do the authors know that it has similar development time and brood size to N2 and JU1400 ?

Other Comments:

Line 53 remove extra spacing

Line 680 correct FISHs to FISH

Line 714 italicize N

Line 983 italicize pathogen name

Some inconsistency throughout in the use of spaces and hyphenating words, especially in materials and methods section. Please check.

Reviewer #3: (No Response)

PLOS authors have the option to publish the peer review history of their article (what does this mean?). If published, this will include your full peer review and any attached files.

Reviewer #1: No

Reviewer #2: No

Reviewer #3: No
---

## [Decision Letter · Decision Letter 1]

20 Feb 2023

Dear %TITLE% Reinke,

We are pleased to inform you that your manuscript 'High-throughput phenotyping of infection by diverse microsporidia species reveals a wild C. elegans strain with opposing resistance and susceptibility traits' has been provisionally accepted for publication in PLOS Pathogens.

Before your manuscript can be formally accepted you will need to complete some formatting changes, which you will receive in a follow up email. A member of our team will be in touch with a set of requests. Please also address the minor comments.

Best regards,

Erik C. Andersen, Ph.D.

Guest Editor

PLOS Pathogens

P'ng Loke

Section Editor

PLOS Pathogens

Kasturi Haldar

Editor-in-Chief

PLOS Pathogens

orcid.org/0000-0001-5065-158X

Michael Malim

Editor-in-Chief

PLOS Pathogens

orcid.org/0000-0002-7699-2064

Reviewer Comments (if any, and for reference):

Reviewer's Responses to Questions

**Part I - Summary**

Reviewer #1: (No Response)

Reviewer #2: The authors have addressed more or less the reviewer comments and the manuscript has improved. I am happy to support publication of this version.

Reviewer #3: The authors have addressed the comments and concerns raised by the reviewers and significantly improved the quality of the manuscript. I have no further questions.

**Part II – Major Issues: Key Experiments Required for Acceptance**

Reviewer #1: None to report

Reviewer #2: (No Response)

Reviewer #3: (No Response)

**Part III – Minor Issues: Editorial and Data Presentation Modifications**

Reviewer #1: Perhaps I missed it, but this sentence in the authors response "We confirmed that 75-87.5% (depending on the Z score cut off) of the strains that we tested individually have the phenotype identified in our PhenoMIP assays" would be a good addition to the manuscript.

In the response to reviews, the authors mention that they did not identify any DEGs with coding variants within the N. Ferruginous susceptibility locus on chromosome I. But it doesn't make much sense to limit the analysis of DEGs to within the susceptibility locus to genes with coding variation because variants that alter the expression of genes can give rise to physiological phenotypes. There is also a discrepancy between what is in the response to reviewers document and what is the main text - response says "genes that have coding variants", the manuscript says "We also analyzed the 12 candidate genes that within in the mapped chromosome I region that contain shared genetic variants in both JU1400 and MY1. None of these genes are differentially regulated in JU1400 upon N. ferruginous infection." Adding more precise language in the manuscript would be ideal.

Reviewer #2: (No Response)

Reviewer #3: (No Response)

PLOS authors have the option to publish the peer review history of their article (what does this mean?). If published, this will include your full peer review and any attached files.

Reviewer #1: No

Reviewer #2: No

Reviewer #3: No

---

## [Editor Report · Acceptance letter]

5 Mar 2023

Dear %TITLE% Reinke,

We are delighted to inform you that your manuscript, "High-throughput phenotyping of infection by diverse microsporidia species reveals a wild *C. elegans* strain with opposing resistance and susceptibility traits," has been formally accepted for publication in PLOS Pathogens.

Best regards,

Kasturi Haldar

Editor-in-Chief

PLOS Pathogens

orcid.org/0000-0001-5065-158X

Michael Malim

Editor-in-Chief

PLOS Pathogens

orcid.org/0000-0002-7699-2064